# Cycle Self-Training for Domain Adaptation

**Hong Liu**
Dept of Electronic Engineering
Tsinghua University
hongliu9903@gmail.com

**Jianmin Wang**
School of Software, BNRist
Tsinghua University
jimwang@tsinghua.edu.cn

**Mingsheng Long**[*]
School of Software, BNRist
Tsinghua University
mingsheng@tsinghua.edu.cn

## Abstract

Mainstream approaches for unsupervised domain adaptation (UDA) learn domain-invariant representations to narrow the domain shift, which are empirically effective but theoretically challenged by the hardness or impossibility theorems. Recently, self-training has been gaining momentum in UDA, which exploits unlabeled target data by training with target pseudo-labels. However, as corroborated in this work, under distributional shift, the pseudo-labels can be unreliable in terms of their large discrepancy from target ground truth. In this paper, we propose *Cycle Self-Training* (CST), a principled self-training algorithm that explicitly enforces pseudo-labels to generalize across domains. CST cycles between a forward step and a reverse step until convergence. In the forward step, CST generates target pseudo-labels with a source-trained classifier. In the reverse step, CST trains a target classifier using target pseudo-labels, and then updates the shared representations to make the target classifier perform well on the source data. We introduce the Tsallis entropy as a confidence-friendly regularization to improve the quality of target pseudo-labels. We analyze CST theoretically under realistic assumptions, and provide hard cases where CST recovers target ground truth, while both invariant feature learning and vanilla self-training fail. Empirical results indicate that CST significantly improves over the state-of-the-arts on visual recognition and sentiment analysis benchmarks.

## 1 Introduction

Transferring knowledge from a source domain with rich supervision to an unlabeled target domain is an important yet challenging problem. Since deep neural networks are known to be sensitive to subtle change in underlying distributions [70], models trained on one labeled dataset often fail to generalize to another unlabeled dataset [58, 1]. Unsupervised domain adaptation (UDA) addresses the challenge of distributional shift by adapting the source model to the unlabeled target data [50, 43].

The mainstream paradigm for UDA is *feature adaptation*, a.k.a. *domain alignment*. By reducing the distance of the source and target feature distributions, these methods learn *invariant representations* to facilitate knowledge transfer between domains [34, 22, 36, 54, 37, 73], with successful applications in various areas such as computer vision [63, 27, 77] and natural language processing [75, 49]. Despite their popularity, the impossibility theories [6] uncovered intrinsic limitations of learning invariant representations when it comes to label shift [74, 32] and shift in the support of domains [29].

Recently, *self-training* (a.k.a. *pseudo-labeling*) [21, 78, 30, 32, 47, 68] has been gaining momentum as a promising alternative to feature adaptation. Originally tailored to semi-supervised learning, self-training generates pseudo-labels of unlabeled data, and jointly trains the model with source labels and target pseudo-labels [31, 39, 30]. However, the *distributional shift* in UDA makes pseudo-labeling more difficult. Directly using all pseudo-labels is risky due to accumulated error and even trivial solution [14]. Thus previous works tailor self-training to UDA by selecting trustworthy pseudo-labels. Using confidence threshold or reweighting, recent works try to alleviate the negative effect of domain

---

[*]Corresponding author: Mingsheng Long (mingsheng@tsinghua.edu.cn)

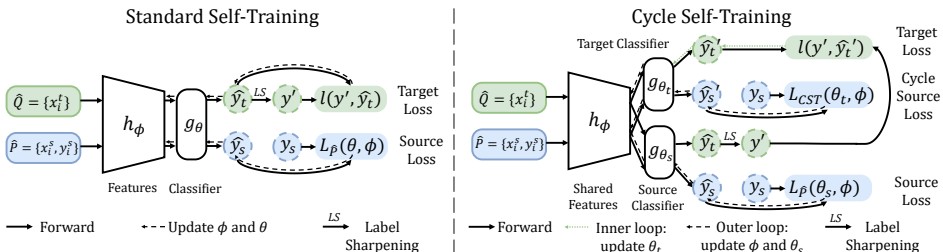

Figure 1: **Standard self-training vs. cycle self-training.** In standard self-training, we generate target pseudo-labels with a source model, and then train the model with both source ground-truths and target pseudo-labels. In cycle self-training, we train a target classifier with target pseudo-labels in the **inner loop**, and make the target classifier perform well on the source domain by updating the shared representations in the **outer loop**.

shift in standard self-training [78, 47], but they can be brittle and require expensive tweaking of the threshold or weight for different tasks, and their performance gain is still inconsistent.

In this work, we first analyze the quality of pseudo-labels with or without domain shift to delve deeper into the difficulty of standard self-training in UDA. On popular benchmark datasets, when the source and target are the same, our analysis indicates that the pseudo-label distribution is almost identical to the ground-truth distribution. However, with distributional shift, their discrepancy can be *very large* with examples of several classes mostly misclassified into other classes. We also study the difficulty of selecting correct pseudo-labels with popular criteria under domain shift. Although entropy and confidence are reasonable selection criteria for correct pseudo-labels without domain shift, the domain shift makes their accuracy decrease sharply.

Our analysis shows that domain shift makes pseudo-labels *unreliable* and that self-training on selected target instances with accurate pseudo-labels is less successful. Thereby, more principled improvement of standard self-training should be tailored to UDA and address the domain shift explicitly. In this work, we propose *Cycle Self-Training* (CST), a principled self-training approach to UDA, which overcomes the limitations of standard self-training (see Figure 1). Different from previous works to select target pseudo-labels with hard-to-tweak protocols, CST learns to generalize the pseudo-labels across domains. Specifically, CST *cycles* between the use of target pseudo-labels to train a target classifier, and the update of shared representations to make the target classifier perform well on the source data. In contrast to the standard Gibbs entropy that makes the target predictions over-confident, we propose a confidence-friendly uncertainty measure based on the *Tsallis entropy* in information theory, which adaptively minimizes the uncertainty without manually tuning or setting thresholds. Our method is simple and generally applicable to vision and language tasks with various backbones.

We empirically evaluate our method on a series of standard UDA benchmarks. Results indicate that CST outperforms previous state-of-the-art methods in 21 out of 25 tasks for object recognition and sentiment classification. Theoretically, we prove that the minimizer of CST objective is endowed with general guarantees of target performance. We also study hard cases on specific distributions, showing that CST recovers target ground-truths while both feature adaptation and standard self-training fail.

## 2 Preliminaries

We study unsupervised domain adaptation (UDA). Consider a source distribution $P$ and a target distribution $Q$ over the input-label space $\mathcal{X} \times \mathcal{Y}$. We have access to $n_s$ labeled *i.i.d.* samples $\widehat{P} = \{x_i^s, y_i^s\}_{i=1}^{n_s}$ from $P$ and $n_t$ unlabeled *i.i.d.* samples $\widehat{Q} = \{x_i^t\}_{i=1}^{n_t}$ from $Q$. The model $f$ comprises a feature extractor $h_\phi$ parametrized by $\phi$ and a head (linear classifier) $g_\theta$ parametrized by $\theta$, i.e. $f_{\theta,\phi}(x) = g_\theta(h_\phi(x))$. The loss function is $\ell(\cdot, \cdot)$. Denote by $L_P(\theta, \phi) := \mathbb{E}_{(x,y) \sim P} \ell(f_{\theta,\phi}(x), y)$ the expected error on $P$. Similarly, we use $L_{\widehat{P}}(\theta, \phi)$ to denote the empirical error on dataset $\widehat{P}$.

We discuss two mainstream UDA methods and their formulations: feature adaptation and self-training.

**Feature Adaptation** trains the model $f$ on the source dataset $\widehat{P}$, and simultaneously matches the source and target distributions in the representation space $\mathcal{Z} = h(\mathcal{X})$:

$$\min_{\theta, \phi} L_{\widehat{P}}(\theta, \phi) + d(h_\sharp \widehat{P}, h_\sharp \widehat{Q}). \tag{1}$$

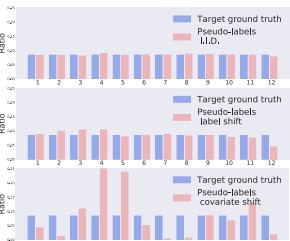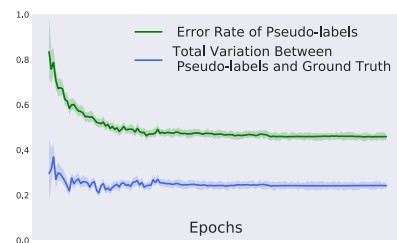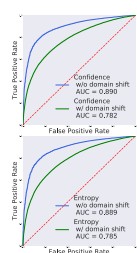

Figure 2: **Analysis of pseudo-labels under domain shift on VisDA-2017.** Left: Pseudo-label distributions with and without domain shift. Middle: Changes of pseudo-label distributions throughout training. Right: Quality of pseudo-labels under different pseudo-label selection criteria.

Here, $h_\sharp \widehat{P}$ denotes the pushforward distribution of $\widehat{P}$, and $d(\cdot, \cdot)$ is some distribution distance. For instance, Long et al. [34] used maximum mean discrepancy $d_{\text{MMD}}$, and Ganin et al. [22] approximated the $\mathcal{H}\Delta\mathcal{H}$-distance $d_{\mathcal{H}\Delta\mathcal{H}}$ [7] with adversarial training. Despite its pervasiveness, recent works have shown the intrinsic limitations of feature adaptation under real-world situations [6, 74, 33, 32, 29].

**Self-Training** is considered a promising alternative to feature adaptation. In this work we mainly focus on pseudo-labeling [31, 30]. Stemming from semi-supervised learning, standard self-training trains a source model $f_s$ on the source dataset $\widehat{P}$: $\min_{\theta_s, \phi_s} L_{\widehat{P}}(\theta_s, \phi_s)$. The *target pseudo-labels* are then generated by $f_s$ on the target dataset $\widehat{Q}$. To leverage unlabeled target data, self-training trains the model on the source and target datasets together with source ground-truths and target pseudo-labels:

$$\min_{\theta, \phi} L_{\widehat{P}}(\theta, \phi) + \mathbb{E}_{x \sim \widehat{Q}} \ell(f_{\theta, \phi}(x), \arg\max_i \{f_{\theta_s, \phi_s}(x)_{[i]}\}). \tag{2}$$

Self-training also uses label-sharpening as a standard protocol [31, 57]. Another popular variant of pseudo-labeling is the teacher-student model [4, 61], which iteratively improves the quality of pseudo-labels via alternatively replacing $\theta_s$ and $\phi_s$ with $\theta$ and $\phi$ of the previous iteration.

## 2.1 Limitations of Standard Self-Training

Standard self-training with pseudo-labels uses unlabeled data efficiently for semi-supervised learning [31, 39, 57]. Here we carry out exploratory studies on the popular VisDA-2017 [45] dataset using ResNet-50 backbones. We find that domain shift makes the pseudo-labels biased towards several classes and thereby unreliable in UDA. See Appendix C.1 for details and results on more datasets.

**Pseudo-label distributions with or without domain shift.** We resample the original VisDA-2017 to simulate different relationship between source and target domains: 1) *i.i.d.*, 2) covariate shift, and 3) label shift. We train the model on the three variants of source dataset and use it to generate target pseudo-labels. We show the distributions of target ground-truths and pseudo-labels in Figure 2 (Left). When the source and target distributions are identical, the distribution of pseudo-labels is almost the same as ground-truths, indicating the reliability of pseudo-labels. In contrast, when exposed to label shift or covariate shift, the distribution of pseudo-labels is significantly different from target ground-truths. Note that classes 2, 7, 8 and 12 appear rarely in the target pseudo-labels in the covariate shift setting, indicating that the pseudo-labels are biased towards several classes due to domain shift. Self-training with these pseudo-labels is risky since it may lead to misalignment of distributions and misclassify many examples of classes 2, 7, 8 and 12.

**Change of pseudo-label distributions throughout training.** To further study the change of pseudo-labels in standard self-training, we compute the total variation (TV) distance between target ground-truths and target pseudo-labels: $d_{\text{TV}}(c, c') = \frac{1}{2} \sum_i \|c_i - c'_i\|$, where $c_i$ is the ratio of class $i$. We plot its change during training in Figure 2 (Middle). Although the error rate of pseudo-labels continues to decrease, $d_{\text{TV}}$ remains almost unchanged at $0.26$ throughout training. Note that $d_{\text{TV}}$ is the lower bound of the error rate of the pseudo-labels (shown in Appendix C.1). If $d_{\text{TV}}$ converges to $0.26$, then the accuracy of pseudo-labels is upper-bounded by $0.74$. This indicates that the important denoising ability [66] of pseudo-labels in standard self-training is hindered by domain shift.

**Difficulty of selecting reliable pseudo-labels under domain shift.** To mitigate the negative effect of false pseudo-labels, recent works proposed to select correct pseudo-labels based on thresholding the entropy or confidence criteria [35, 21, 37, 57]. However, it remains unclear whether these strategies are still effective under domain shift. Here we compare the quality of pseudo-labels selected by

different strategies with or without domain shift. For each strategy, we compute False Positive Rate and True Positive Rate for different thresholds and plot its ROC curve in Figure 2 (Right). When the source and target distributions are identical, both entropy and confidence are reasonable strategies for selecting correct pseudo-labels (AUC=0.89). However, when the target pseudo-labels are generated by the source model, the quality of pseudo-labels decreases sharply under domain shift (AUC=0.78).

## 3 Approach

We present Cycle Self-Training (CST) to improve pseudo-labels under domain shift. An overview of our method is given in Figure 1. Cycle Self-Training iterates between a *forward step* and a *reverse step* to make self-trained classifiers generalize well on both target and source domains.

### 3.1 Cycle Self-Training

**Forward Step.** Similar to standard self-training, we have a source classifier $\theta_s$ trained on top of the shared representations $\phi$ on the labeled source domain, and use it to generate target pseudo-labels as

$$y' = \arg\max_i \{f_{\theta_s,\phi}(x)_{[i]}\}, \tag{3}$$

for each $x$ in the target dataset $\widehat{Q}$. Traditional self-training methods use confidence thresholding or reweighting to select reliable pseudo-labels. For example, Sohn et al. [57] select pseudo-labels with softmax value and Long et al. [37] add entropy reweighting to rely on examples with more confidence prediction. However, the output of deep networks is usually miscalibrated [25], and is not necessarily related to the ground-truth confidence even on the same distribution. In domain adaptation, as shown in Section 2.1, the discrepancy between the source and target domains makes pseudo-labels even more unreliable, and the performance of commonly used selection strategies is also unsatisfactory. Another drawback is the expensive tweaking in order to find the optimal confidence threshold for new tasks. To better apply self-training to domain adaptation, we expect that the model can gradually refine the pseudo-labels by itself without the cumbersome selection or thresholding.

**Reverse Step.** We design a complementary step with the following insights to improve self-training. Intuitively, the labels on the source domain contain both useful information that can transfer to the target domain and harmful information that can make pseudo-labels incorrect. Similarly, *reliable pseudo-labels* on the target domain can transfer to the source domain in turn, while models trained with incorrect pseudo-labels on the target domain cannot transfer to the source domain. In this sense, if we explicitly train the model to make target pseudo-labels informative of the source domain, we can gradually make the pseudo-labels more accurate and learn to generalize to the target domain.

Specifically, with the pseudo-labels $y'$ generated by the source classifier $\theta_s$ at hand as in equation 3, we train a target head $\hat{\theta}_t(\phi)$ on top of the representation $\phi$ with pseudo-labels on the target domain $\widehat{Q}$,

$$\hat{\theta}_t(\phi) = \arg\min_\theta \mathbb{E}_{x \sim \widehat{Q}} \ell(f_{\theta,\phi}(x), y'). \tag{4}$$

We wish to make the target pseudo-labels informative of the source domain and gradually refine them. To this end, we update the shared feature extractor $\phi$ to predict accurately on the source domain and jointly *enforce the target classifier $\hat{\theta}_t(\phi)$ to perform well on the source domain*. This naturally leads to the objective of **Cycle Self-Training**:

$$\min_{\theta_s,\phi} L_{\text{Cycle}}(\theta_s, \phi) := L_{\widehat{P}}(\theta_s, \phi) + L_{\widehat{P}}(\hat{\theta}_t(\phi), \phi). \tag{5}$$

**Bi-level Optimization.** The objective in equation 5 relies on the solution $\hat{\theta}_t(\phi)$ to the objective in equation 4. Thus, CST formulates a *bi-level* optimization problem. In the **inner loop** we generate target pseudo-labels with the source classifier (equation 3), and train a target classifier with target pseudo-labels (equation 4). After each inner loop, we update the feature extractor $\phi$ for one step in the **outer loop** (equation 5), and start a new inner loop again. However, since the inner loop of the optimization in equation 4 only involves the light-weight linear head $\theta_t$, we propose to calculate the analytical form of $\hat{\theta}_t(\phi)$ and directly back-propagate to the feature extractor $\phi$ instead of calculating the second-order derivatives as in MAML [18]. The resulting framework is as fast as training two heads jointly. Also note that the solution $\hat{\theta}_t(\phi)$ relies on $\theta_s$ implicitly through $y'$. However, both standard self-training and our implementation use label sharpening, making $y'$ not differentiable. Thus we follow vanilla self-training and *do not* consider the gradient of $\hat{\theta}_t(\phi)$ w.r.t. $y'$ in the outer loop optimization. We defer the derivation and implementation of bi-level optimization to Appendix B.2.

### 3.2 Tsallis Entropy Minimization

Gibbs entropy is widely used by existing semi-supervised learning methods to regularize the model output and minimize the uncertainty of predictions on unlabeled data [24]. In this work, we generalize Gibbs entropy to Tsallis entropy [62] in information theory. Suppose the softmax output of a model is $y \in \mathbb{R}^K$, then the $\alpha$-*Tsallis entropy* is defined as

$$S_\alpha(y) = \frac{1}{\alpha - 1} \left( 1 - \sum y_{[i]}^\alpha \right), \tag{6}$$

where $\alpha > 0$ is the *entropic-index*. Note that $\lim_{\alpha \to 1} S_\alpha(y) = \sum_i -y_{[i]} \log(y_{[i]})$ which exactly recovers the Gibbs entropy. When $\alpha = 2$, $S_\alpha(y)$ becomes the Gini impurity $1 - \sum_i y_{[i]}^2$.

We propose to control the uncertainty of target pseudo-labels based on **Tsallis entropy minimization**:

$$L_{\widehat{Q},\text{Tsallis},\alpha}(\theta, \phi) := \mathbb{E}_{x \sim \widehat{Q}} S_\alpha(f_{\theta,\phi}(x)). \tag{9}$$

Figure 3 shows the change of Tsallis entropy with different entropic-indices $\alpha$ for binary problems. Intuitively, smaller $\alpha$ exerts more penalization on uncertain predictions and larger $\alpha$ allows several scores $y_i$'s to be similar. This is critical in self-training since an overly small $\alpha$ (as in Gibbs entropy) will make the incorrect dimension of pseudo-labels close to 1 and have no chance to be corrected throughout training. In Section 5.4, we further verify this property with experiments.

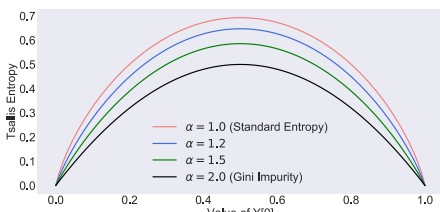

Figure 3: Tsallis entropy vs. entropic-index $\alpha$.

An important improvement of the Tsallis entropy over Gibbs entropy is that it can choose the suitable measure of uncertainty for different systems to avoid over-confidence caused by overly penalizing the uncertain pseudo-labels. To automatically find the suitable $\alpha$, we adopt a similar strategy as Section 3.1. The intuition is that if we use the suitable entropic-index $\alpha$ to train the source classifier $\theta_{s,\alpha}$, the target pseudo-labels generated by $\theta_{s,\alpha}$ will contain desirable knowledge of the source dataset, i.e. a target classifier $\theta_{t,\alpha}$ trained with these pseudo-labels will perform well on the source domain. Therefore, we semi-supervisedly train a classifier $\hat{\theta}_{s,\alpha}$ on the source domain with the $\alpha$-Tsallis entropy regularization $L_{\widehat{Q},\text{Tsallis},\alpha}$ on the target domain as: $\hat{\theta}_{s,\alpha} = \arg \min_\theta L_{\widehat{P}}(\theta, \phi) + L_{\widehat{Q},\text{Tsallis},\alpha}(\theta, \phi)$, from which we obtain the target pseudo-labels. Then we train another head $\hat{\theta}_{t,\alpha}$ with target pseudo-labels. We automatically find $\alpha$ by minimizing the loss of $\hat{\theta}_{t,\alpha}$ on the source data:

$$\hat{\alpha} = \underset{\alpha \in [1,2]}{\arg \min} \, L_{\widehat{P}}(\hat{\theta}_{t,\alpha}, \phi) \tag{10}$$

To solve equation 10, we discretize the feasible region $[1, 2]$ of $\alpha$ and use discrete optimization to lower computational cost. We also update $\alpha$ at the start of each epoch, since we found more frequent

---

**Algorithm 1** Cycle Self-Training (CST)

1: **Input:** source dataset $\widehat{P}$ and target dataset $\widehat{Q}$.
2: **for** epoch $= 0$ **to** MaxEpoch **do**
3:     Select $\hat{\alpha}$ as equation 10 at the start of each epoch.
4:     **for** $t = 0$ **to** MaxIter **do**
5:         **Forward Step**
6:         Generate pseudo-labels on the target domain with $\phi$ and $\theta_s$: $y' = \arg \max_i \{ f_{\theta_s,\phi}(x)_{[i]} \}$.
7:         **Reverse Step**
8:         Train a target head $\hat{\theta}_t(\phi)$ with target pseudo-labels $y'$ on the feature extractor $\phi$:

$$\hat{\theta}_t(\phi) = \underset{\theta}{\arg \min} \, \mathbb{E}_{x \sim \widehat{Q}} \ell(f_{\theta,\phi}(x), y').$$

9:         Update the feature extractor $\phi$ and the source head $\theta_s$ to make $\hat{\theta}_t(\phi)$ perform well on the source dataset and minimize the $\hat{\alpha}$-Tsallis entropy on the target dataset:

$$\phi \leftarrow \phi - \eta \nabla_\phi [L_{\widehat{P}}(\theta_s, \phi) + L_{\widehat{P}}(\hat{\theta}_t(\phi), \phi) + L_{\widehat{Q},\text{Tsallis},\hat{\alpha}}(\theta_s, \phi)]. \tag{7}$$

$$\theta_s \leftarrow \theta_s - \eta \nabla_{\theta_s} [L_{\widehat{P}}(\theta_s, \phi) + L_{\widehat{Q},\text{Tsallis},\hat{\alpha}}(\theta_s, \phi)]. \tag{8}$$

10:     **end for**
11: **end for**

---

update leads to no performance gain. Details are deferred to Appendix B.3. Finally, with the optimal $\hat{\alpha}$ found, we add the $\hat{\alpha}$-Tsallis entropy minimization term $L_{\widehat{Q},\text{Tsallis},\hat{\alpha}}$ to the overall objective:

$$\underset{\theta_s,\phi}{\text{minimize}}\, L_{\text{Cycle}}(\theta_s,\phi) + L_{\widehat{Q},\text{Tsallis},\hat{\alpha}}(\theta_s,\phi). \tag{11}$$

In summary, Algorithm 1 depicts the complete training procedure of Cycle Self-Training (CST).

## 4 Theoretical Analysis

We analyze the properties of CST theoretically. First, we prove that the minimizer of the CST loss $L_{\text{CST}}(f_s, f_t)$ will lead to small target loss $\text{Err}_Q(f_s)$ under a simple but realistic expansion assumption. Then, we further demonstrate a concrete instantiation where cycle self-training provably recovers the target ground truth, but both feature adaptation and standard self-training fail. *Due to space limit, we state the main results here and defer all proof details to Appendix A.*

### 4.1 CST Provably Works under the Expansion Assumption

We start from a $K$-way classification model, $f : \mathcal{X} \to [0,1]^K \in \mathcal{F}$ and $\tilde{f}(x) := \arg\max_i f(x)_{[i]}$ denotes the prediction. Denote by $P_i$ the conditional distribution of $P$ given $y = i$. Assume the supports of $P_i$ and $P_j$ are disjoint for $i \neq j$. The definition is similar for $Q_i$. We further Assume $P(y = i) = Q(y = i)$. For any $x \in \mathcal{X}$, $\mathcal{N}(x)$ is defined as the *neighboring set* of $x$ with a proper metric $d(\cdot,\cdot)$, $\mathcal{N}(x) = \{x' : d(x,x') \leq \xi\}$. $\mathcal{N}(A) := \cup_{x\in A}\mathcal{N}(x)$. Denote the expected error on the target domain by $\text{Err}_Q(f) := \mathbb{E}_{(x,y)\sim Q}\mathbb{I}(\tilde{f}(x) \neq y)$.

We study the CST algorithm under the *expansion assumption* of the mixture distribution [66, 11]. Intuitively, this assumption indicates that the conditional distributions $P_i$ and $Q_i$ are closely located and regularly shaped, enabling knowledge transfer from the source domain to the target domain.

**Definition 1** (($q,\epsilon$)-**constant expansion** [66]). *We say $P$ and $Q$ satisfy $(q,\epsilon)$-constant expansion for some constant $q, \epsilon \in (0,1)$, if for any set $A \in \mathcal{X}$ and any $i \in [K]$ with $\frac{1}{2} > P_{\frac{1}{2}(P_i+Q_i)}(A) > q$, we have $P_{\frac{1}{2}(P_i+Q_i)}(\mathcal{N}(A)\backslash A) > \min\{\epsilon, P_{\frac{1}{2}(P_i+Q_i)}(A)\}$.*

Based on this expansion assumption, we consider a *robustness-constrained* version of CST. Later we will show that the robustness is closely related to the uncertainty. Denote by $f_s$ the source model and $f_t$ the model trained on the target with pseudo-labels. Let $R(f_t) := P_{\frac{1}{2}(P+Q)}(\{x : \exists x' \in \mathcal{N}(x), \tilde{f}_t(x) \neq \tilde{f}_t(x')\})$ represent the robustness [66] of $f_t$ on $P$ and $Q$. Suppose $\mathbb{E}_{(x,y)\sim Q}\mathbb{I}(\tilde{f}_s(x) \neq \tilde{f}_t(x)) \leq c$ and $R(f_t) \leq \rho$. The following theorem states that when $f_s$ and $f_t$ behave similarly on the target domain $Q$ and $f_t$ is robust to local changes in input, the minimizer of the cycle source error $\text{Err}_P(f_t)$ will guarantee low error of $f_s$ on the target domain $Q$.

**Theorem 1.** *Suppose Definition 1 holds for $P$ and $Q$. For any $f_s, f_t$ satisfying $\mathbb{E}_{(x,y)\sim Q}\mathbb{I}(\tilde{f}_s(x) \neq \tilde{f}_t(x)) \leq c$ and $R(f_t) \leq \rho$, the expected error of $f_s$ on the target domain $Q$ is bounded,*

$$\text{Err}_Q(f_s) \leq \text{Err}_P(f_t) + c + 2q + \frac{\rho}{\min\{\epsilon,q\}}. \tag{12}$$

To further relate the expected error with the CST training objective and obtain finite-sample guarantee, we use the multi-class margin loss: $l_\gamma(f(x), y) := \psi_\gamma(-\mathcal{M}(f(x), y))$, where $\mathcal{M}(v, y) = v_{[y]} - \max_{y'\neq y} v_{[y']}$ and $\psi_\gamma$ is the ramp function. We then extend the margin loss: $\mathcal{M}(v) = \max_y(v_{[y]} - \max_{y'\neq y} v_{[y']})$ (The difference between the largest and the second largest scores in $v$), and $l_\gamma(f_t(x), f_s(x)) := \psi_\gamma(-\mathcal{M}(f_t(x), \tilde{f}_s(x)))$. Further suppose $f_{[i]}$ is $L_f$-Lipschitz w.r.t. the metric $d(\cdot,\cdot)$ and $\tau := 1 - 2L_f\xi\min\{\epsilon, q\} > 0$. Consider the following training objective for CST, denoted by $L_{\text{CST}}(f_s, f_t)$, where $L_{\widehat{P},\gamma}(f_t) := \mathbb{E}_{(x,y)\sim\widehat{P}}l_\gamma(f_t(x), y)$ corresponds to the cycle source loss in equation 5, $L_{\widehat{Q},\gamma}(f_t, f_s) := \mathbb{E}_{(x,y)\sim\widehat{Q}}l_\gamma(f_t(x), f_s(x))$ is consistent with the target loss in equation 4, and $\mathcal{M}(f_t(x))$ is closely related to the uncertainty of predictions in equation 11.

$$\min L_{\text{CST}}(f_s, f_t) := L_{\widehat{P},\gamma}(f_t) + L_{\widehat{Q},\gamma}(f_t, f_s) + \frac{1 - \mathbb{E}_{(x,y)\sim\frac{1}{2}(\widehat{P}+\widehat{Q})}\mathcal{M}(f_t(x))}{\tau}. \tag{13}$$

The following theorem shows that the minimizer of the training objective $L_{\text{CST}}(f_s, f_t)$ guarantees low population error of $f_s$ on the target domain $Q$.

**Theorem 2.** $\widehat{\mathcal{R}}(\mathcal{F}|_{\widehat{P}})$ *denotes the empirical Rademacher complexity of function class $\mathcal{F}$ on dataset $\widehat{P}$. For any solution of equation 13 and $\gamma > 0$, with probability larger than $1 - \delta$,*

$$\text{Err}_Q(f_s) \leq L_{\text{CST}}(f_s, f_t) + 2q + \frac{4K}{\gamma}\left[\widehat{\mathcal{R}}(\mathcal{F}|_{\widehat{P}}) + \widehat{\mathcal{R}}(\tilde{\mathcal{F}} \times \mathcal{F}|_{\widehat{Q}})\right] + \frac{2}{\tau}\left[\widehat{\mathcal{R}}(\mathcal{F}|_{\widehat{P}}) + \widehat{\mathcal{R}}(\mathcal{F}|_{\widehat{Q}})\right] + \zeta,$$

*where $\zeta = O\left(\sqrt{\log(1/\delta)/n_s} + \sqrt{\log(1/\delta)/n_t}\right)$ is a low-order term. $\tilde{\mathcal{F}} \times \mathcal{F}$ refers to the function class $\{x \to f(x)_{[\tilde{f}'(x)]} : f, f' \in \mathcal{F}\}$.*

**Main insights.** Theorem 2 justifies CST under the expansion assumption. The generalization error of the classifier $f_s$ on the target domain is bounded with the CST loss objective $L_{\text{CST}}(f_s, f_t)$, the intrinsic property of the data distribution $q$, and the complexity of the function classes. In our algorithm, $L_{\text{CST}}(f_s, f_t)$ is minimized by the neural networks and $q$ is a constant. The complexity of the function class can be controlled with proper regularization.

## 4.2 Hard Case for Feature Adaptation and Standard Self-Training

To gain more insight, we study UDA in a quadratic neural network $f_{\theta,\phi}(x) = \theta^\top(\phi^\top x)^{\odot 2}$, where $\odot$ is element-wise power. In UDA, the source can have *multiple solutions* but we aim to learn the one working on the target [34]. We design the underlying distributions $p$ and $q$ in Table 6 to reflect this. Consider the following $P$ and $Q$. $x_{[1]}$ and $x_{[2]}$ are sampled *i.i.d.* from distribution $p$ on $P$, and from

q on Q. For $i \in [3, d]$, $x_{[i]} = \sigma_i x_{[2]}$ on $P$ and $x_{[i]} = \sigma_i x_{[1]}$ on $Q$. $\sigma_i \in \{\pm 1\}$ are *i.i.d.* and uniform. We also assume realizability: $y = x_{[1]}^2 - x_{[2]}^2$ for both source and target. Note that $y = x_{[1]}^2 - x_{[i]}^2$ for all $i \in [2, d]$ are solutions to $P$ but only $y = x_{[1]}^2 - x_{[2]}^2$ works on $Q$. We visualize this specialized setting in Figure 4.

Table 1: The design of $p$ and $q$.

| Distribution | $-1$ | $+1$ | $0$ |
|---|---|---|---|
| Source $p$ | 0.05 | 0.05 | 0.90 |
| Target $q$ | 0.25 | 0.25 | 0.50 |

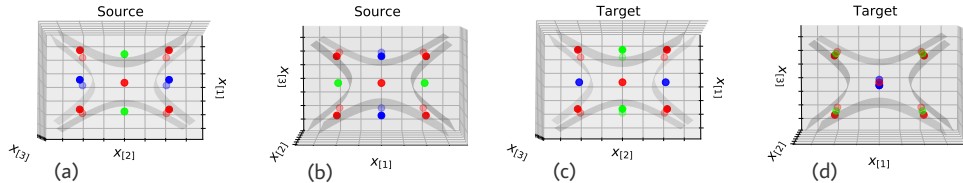

(a)         (b)         (c)         (d)

Figure 4: **The hard case where $d = 3$.** Green dots for $y = 1$, red dots for $y = 0$, and blue dots for $y = -1$. The grey curve is the classification boundary of different features. The good feature $x_{[1]}^2 - x_{[2]}^2$ works on the target domain (shown in (a) and (c)), whereas the spurious feature $x_{[1]}^2 - x_{[3]}^2$ only works on the source domain (shown in (b) and (d)). In Section 4.2, we show that feature adaptation and standard self-training learn $x_{[1]}^2 - x_{[3]}^2$, while CST learns $x_{[1]}^2 - x_{[2]}^2$.

To make the features more tractable, we study the norm-constrained version of the algorithms (details are deferred to Section A.3.2). We compare the features learned by feature adaptation, standard self-training, and CST. Intuitively, feature adaptation fails because the *ideal* target solution $y = x_{[1]}^2 - x_{[2]}^2$ has larger distance in the feature space than other spurious solutions $y = x_{[1]}^2 - x_{[i]}^2$. Standard self-training also fails since it will choose randomly among all solutions. In comparison, CST can recover the ground truth, because it can distinguish the spurious solution resulting in *bad pseudo-labels*. A classifier trained with those pseudo-labels *cannot* work on the source domain in turn. This intuition is rigorously justified in the following two theorems.

**Theorem 3.** *For $\epsilon \in (0, 0.5)$, the following statements hold for feature adaptation and self-training:*

- *For failure rate $\xi > 0$, and target dataset size $n_t > \Theta(\log\frac{1}{\xi})$, with probability at least $1 - \xi$ over the sampling of target data, the solution $(\hat{\theta}_{\text{FA}}, \hat{\phi}_{\text{FA}})$ found by feature adaptation satisfies*

$$\text{Err}_Q(\hat{\theta}_{\text{FA}}, \hat{\phi}_{\text{FA}}) \geq \epsilon. \tag{14}$$

- *With probability at least $1 - \frac{1}{d-1}$, the solution $(\hat{\theta}_{\text{ST}}, \hat{\phi}_{\text{ST}})$ of standard self-training satisfies*

$$\text{Err}_Q(\hat{\theta}_{\text{ST}}, \hat{\phi}_{\text{ST}}) \geq \epsilon. \tag{15}$$

**Theorem 4.** *For failure rate $\xi > 0$, and target dataset size $n_t > \Theta(\log \frac{1}{\xi})$, with probability at least $1 - \xi$, the solution of CST $(\hat{\phi}_{\text{CST}}, \hat{\theta}_{\text{CST}})$ recovers the ground truth of the target dataset:*

$$\text{Err}_Q(\hat{\theta}_{\text{CST}}, \hat{\phi}_{\text{CST}}) = 0. \tag{16}$$

## 5 Experiments

We test the performance of the proposed method on both vision and language datasets. Cycle Self-Training (CST) consistently outperforms state-of-the-art feature adaptation and self-training methods. Code is available at https://github.com/Liuhong99/CST.

### 5.1 Setup

**Datasets.** We experiment on visual object recognition and linguistic sentiment classification tasks: *Office-Home* [64] has 65 classes from four kinds of environment with large domain gap: *Artistic* (**Ar**), *Clip Art* (**Cl**), *Product* (**Pr**), and *Real-World* (**Rw**); *VisDA-2017* [45] is a large-scale UDA dataset with two domains named **Synthetic** and **Real**. The datasets consist of over 200k images from 12 categories of objects; *Amazon Review* [10] is a linguistic sentiment classification dataset of product reviews in four products: *Books* (**B**), *DVDs* (**D**), *Electronics* (**E**), and *Kitchen* (**K**).

**Implementation.** We use **ResNet-50** [26] (pretrained on ImageNet [53]) as feature extractors for vision tasks, and **BERT** [16] for linguistic tasks. On VisDA-2017, we also provide results of ResNet-101 to include more baselines. We use cross-entropy loss for classification on the source domain. When training the target head $\hat{\theta}_t$ and updating the feature extractor with CST, we use squared loss to get the analytical solution of $\hat{\theta}_t$ directly and avoid calculating second order derivatives as meta-learning [18]. Details on adapting squared loss to multi-class classification are deferred to Appendix B. We adopt SGD with initial learning rate $\eta_0 = 2e - 3$ for image classification and $\eta_0 = 5e - 4$ for sentiment classification. Following standard protocol in [26], we decay the learning rate by $0.1$ each 50 epochs until 150 epochs. We run all the tasks 3 times and report mean and deviation in top-1 accuracy. For VisDA-2017, we report the mean class accuracy. Following Theorem 2, we also enhance CST with sharpness-aware regularization [19] (CST+SAM), which help regularize the Lipschitzness of the function class. Due to space limit, we report mean accuracies in Tables 2 and 3 and defer standard deviation to Appendix C.

### 5.2 Baselines

We compare with two lines of works in domain adaptation: feature adaptation and self-training. We also compare with more complex state-of-the-arts and create stronger baselines by combining feature adaptation and self-training.

**Feature Adaptation:** DANN [22], MCD [54], CDAN [37] (which improves DANN with pseudo-label conditioning), MDD [73] (which improves previous domain adaptation with margin theory), Implicit Alignment (IA) [28] (which improves MDD to deal with label shift).

**Self-Training.** We include VAT [40], MixMatch [8] and FixMatch [57] in the semi-supervised learning literature as self-training methods. We also compare with self-training methods for UDA: CBST [77], which considers class imbalance in standard self-training, and KLD [78], which improves CBST with label regularization. However, these methods involve tricks specified for convolutional networks. Thus, in sentiment classification tasks where we use BERT backbones, we compare with other consistency regularization baselines: VAT [40], VAT+Entropy Minimization.

**Feature Adaptation + Self-Training.** DIRT-T [56] combines DANN, VAT, and entropy minimization. We also create more powerful baselines: CDAN+VAT+Entropy and MDD+Fixmatch.

**Other SOTA.** AFN [69] boosts transferability by large norm. STAR [38] aligns domains with stochastic classifiers. SENTRY [48] selects confident examples with a committee of random augmentations.

### 5.3 Results

Results on 12 pairs of *Office-Home* tasks are shown in Table 2. When domain shift is large, standard self-training methods such as VAT and FixMatch suffer from the decay in pseudo-label quality. **CST** outperforms feature adaptation and self-training methods significantly in 9 out of 12 tasks. Note that CST does not involve manually setting confidence threshold or reweighting.

Table 2: Accuracy (%) on Office-Home for unsupervised domain adaptation (`ResNet-50`).

| Method | Ar-Cl | Ar-Pr | Ar-Rw | Cl-Ar | Cl-Pr | Cl-Rw | Pr-Ar | Pr-Cl | Pr-Rw | Rw-Ar | Rw-Cl | Rw-Pr | Avg. |
|---|---|---|---|---|---|---|---|---|---|---|---|---|---|
| DANN [22] | 45.6 | 59.3 | 70.1 | 47.0 | 58.5 | 60.9 | 46.1 | 43.7 | 68.5 | 63.2 | 51.8 | 76.8 | 57.6 |
| CDAN [37] | 50.7 | 70.6 | 76.0 | 57.6 | 70.0 | 70.0 | 57.4 | 50.9 | 77.3 | 70.9 | 56.7 | 81.6 | 65.8 |
| CDAN+VAT+Entropy | 52.2 | 71.5 | 76.4 | 61.1 | 70.3 | 67.8 | 59.5 | 54.4 | 78.6 | 73.2 | 59.0 | 82.7 | 67.3 |
| FixMatch [57] | 51.8 | 74.2 | 80.1 | 63.5 | 73.8 | 61.3 | 64.7 | 51.4 | 80.0 | 73.3 | 56.8 | 81.7 | 67.7 |
| MDD [73] | 54.9 | 73.7 | 77.8 | 60.0 | 71.4 | 71.8 | 61.2 | 53.6 | 78.1 | 72.5 | 60.2 | 82.3 | 68.1 |
| MDD+IA [28] | 56.2 | 77.9 | 79.2 | 64.4 | 73.1 | 74.4 | 64.2 | 54.2 | 79.9 | 71.2 | 58.1 | 83.1 | 69.5 |
| SENTRY [48] | **61.8** | 77.4 | 80.1 | 66.3 | 71.6 | 74.7 | 66.8 | **63.0** | 80.9 | 74.0 | **66.3** | 84.1 | 72.2 |
| **CST** | 59.0 | **79.6** | **83.4** | **68.4** | **77.1** | **76.7** | **68.9** | 56.4 | **83.0** | **75.3** | 62.2 | **85.1** | **73.0** |

Table 3: Accuracy (%) on Multi-Domain Sentiment Dataset for domain adaptation with `BERT`.

| Method | B-D | B-E | B-K | D-B | D-E | D-K | E-B | E-D | E-K | K-B | K-D | K-E | Avg. |
|---|---|---|---|---|---|---|---|---|---|---|---|---|---|
| Source-only | 89.7 | 88.4 | 90.9 | 90.1 | 88.5 | 90.2 | 86.9 | 88.5 | 91.5 | 87.6 | 87.3 | 91.2 | 89.2 |
| DANN [22] | 90.2 | 89.5 | 90.9 | 91.0 | 90.6 | 90.2 | 87.1 | 87.5 | 92.8 | 87.8 | 87.6 | 93.2 | 89.9 |
| VAT [40] | 90.6 | 91.0 | 91.7 | 90.8 | 90.8 | 92.0 | 87.2 | 86.9 | 92.6 | 86.9 | 87.7 | 92.9 | 90.1 |
| VAT+Entropy | 90.4 | 91.3 | 91.5 | 91.0 | 91.1 | 92.4 | 87.5 | 86.3 | 92.4 | 86.5 | 87.5 | 93.1 | 90.1 |
| MDD [73] | 90.4 | 90.4 | 91.8 | 90.2 | 90.9 | 91.0 | 87.5 | 86.3 | 92.5 | 89.0 | 87.9 | 92.1 | 90.0 |
| **CST** | **91.5** | **92.9** | **92.6** | **91.9** | **92.6** | **93.5** | **90.2** | **89.4** | **93.8** | 87.9 | **88.3** | **93.5** | **91.5** |

Table 4 shows the results on *VisDA-2017*. **CST** surpasses state-of-the-arts with ResNet-50 and ResNet-101 backbones. We also combine feature adaptation and self-training (DIRT-T, CDAN+VAT+entropy and MDD+FixMatch) to test if feature adaptation alleviates the negative effect of domain shift in standard self-training. Results indicate that CST is a better solution than simple combination.

While most traditional self-training methods include techniques specified for ConvNets such as Mixup [72], **CST** is a *universal* method and can directly work on sentiment classification by simply replacing the head and training objective of BERT [16]. In Table 3, most feature adaptation baselines improve over source only marginally, but **CST** outperforms all baselines on most tasks significantly.

### 5.4 Analysis

**Ablation Study.** We study the role of each part of CST in self-training. CST w/o Tsallis removes the Tsallis entropy $L_{\text{Tsallis},\alpha}$. CST+Entropy replaces the Tsallis entropy with standard entropy. FixMatch+Tsallis adds $L_{\text{Tsallis},\alpha}$ to standard self-training. Observations are shown in Table 5. CST+Entropy performs 3.7% worse than CST, indicating that Tsallis entropy is a better regularization for pseudo-labels than standard entropy. CST performs 5.4% better than FixMatch, indicating that CST is better adapted to domain shift than standard self-training. While FixMatch+Tsallis outperforms FixMatch, it is still 3.6% behind CST, with much larger total variation distance $d_{\text{TV}}$ between pseudo-labels and ground-truths, indicating that CST makes pseudo-labels more reliable than standard self-training under domain shift.

Table 5: Ablation on VisDA-2017.

| Method | Accuracy ↑ | $d_{\text{TV}}$ ↓ |
|---|---|---|
| FixMatch [57] | $74.5 \pm 0.2$ | 0.22 |
| Fixmatch+Tsallis | $76.3 \pm 0.8$ | 0.15 |
| CST w/o Tsallis | $72.0 \pm 0.4$ | 0.16 |
| CST+Entropy | $76.2 \pm 0.6$ | 0.20 |
| **CST** | $\mathbf{79.9 \pm 0.5}$ | 0.12 |

**Quality of Pseudo-labels.** We visualize the error of pseudo-labels during training on VisDA-2017 in Figure 5 (Left). The error of target classifier $\theta_t$ on the source domain decreases quickly in training, when both the error of pseudo-labels (error of $\theta_s$ on $Q$) and the total variation (TV) distance between pseudo-labels and ground-truths continue to decay, indicating that CST gradually refines pseudo-labels. This forms a clear contrast to standard self-training as visualized in Figure 2 (Middle), where the distance $d_{\text{TV}}$ remains nearly unchanged throughout training.

**Comparison of Gibbs entropy and Tsallis entropy.** We compare the pseudo-labels learned with standard Gibbs entropy and Tsallis entropy on Ar→Cl with ResNet-50 at epoch 40. We compute the difference between *the largest and the second largest softmax scores* of each target example and plot the histogram in Figure 5 (Right). Gibbs entropy makes the largest softmax output close to 1, indicating over-confidence. In this case, if the prediction is wrong, it can be hard to correct it using self-training. In contrast, Tsallis entropy allows the largest and the second largest scores to be similar.

Table 4: Mean Class Accuracy (%) for unsupervised domain adaptation on VisDA-2017.

| Method | ResNet-50 | ResNet-101 | Method | ResNet-50 | ResNet-101 |
|---|---|---|---|---|---|
| DANN [22] | 69.3 | 79.5 | CBST [77] | – | $76.4 \pm 0.9$ |
| VAT [40] | $68.0 \pm 0.3$ | $73.4 \pm 0.5$ | KLD [78] | – | $78.1 \pm 0.2$ |
| DIRT-T [56] | $68.2 \pm 0.3$ | $77.2 \pm 0.5$ | MDD [73] | 74.6 | $81.6 \pm 0.3$ |
| MCD [54] | 69.2 | 77.7 | AFN [69] | – | 76.1 |
| CDAN [37] | 70.0 | 80.1 | MDD+IA [28] | 75.8 | – |
| CDAN+VAT+Entropy | $76.5 \pm 0.5$ | $80.4 \pm 0.7$ | MDD+FixMatch | $77.8 \pm 0.3$ | $82.4 \pm 0.4$ |
| MixMatch | $69.3 \pm 0.4$ | $77.0 \pm 0.5$ | STAR [38] | – | 82.7 |
| FixMatch [57] | $74.5 \pm 0.2$ | $79.5 \pm 0.3$ | SENTRY [48] | 76.7 | – |
| **CST** | $\underline{79.9 \pm 0.5}$ | $\underline{84.8 \pm 0.6}$ | **CST+SAM** | $\mathbf{80.6 \pm 0.5}$ | $\mathbf{86.5 \pm 0.7}$ |

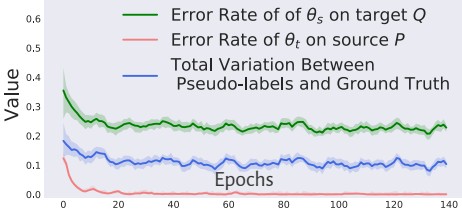 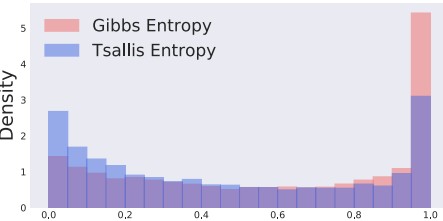

Figure 5: **Analysis.** Left: Error of pseudo-labels and reverse pseudo-labels. The error of target classifier $\theta_t$ on the source domain decreases, indicating the quality of pseudo-labels is refined. Right: Histograms of the difference between the largest and the second largest softmax scores. Tsallis entropy avoids over-confidence.

## 6 Related Work

**Self-Training.** Self-training is a mainstream technique for semi-supervised learning [13]. In this work, we focus on pseudo-labeling [52, 31, 2], which uses unlabeled data by training on pseudo-labels generated by a source model. Other lines of work study consistency regularization [4, 51, 55, 40]. Recent works demonstrate the power of such methods [67, 57, 23]. Equipped with proper training techniques, these methods can achieve comparable results as standard training that uses much more labeled examples [17]. Zoph et al. [76] compare self-training to pre-training and joint training. Vu et al. [65], Mukherjee & Awadallah [42] show that task-level self-training works well in few-shot learning. These methods are tailored to semi-supervised learning or general representation learning and do not take domain shift into consideration explicitly. Wei et al. [66], Frei et al. [20] provide the first nice theoretical analysis of self-training based on the expansion assumption.

**Domain Adaptation.** Inspired by the generalization error bound of Ben-David et al. [7], Long et al. [34], Zellinger et al. [71] minimize distance measures between source and target distributions to learn domain-invariant features. Ganin et al. [22] (DANN) proposed to approximate the domain distance by adversarial learning. Follow-up works proposed various improvement upon DANN [63, 54, 37, 73, 28]. Popular as they are, failure cases exist in situation like label shift [74, 32], shift in support of domains [29], and large discrepancy between source and target [33]. Another line of works try to address domain adaptation with self-training. Shu et al. [56] improves DANN with VAT and entropy minimization. French et al. [21], Zou et al. [78], Li et al. [32] incorporated various semi-supervised learning techniques to boost domain adaptation performance. Kumar et al. [30], Chen et al. [15] and Cai et al. [11] showed self-training provably works in domain adaptation under certain assumptions.

## 7 Conclusion

We propose cycle self-training in place of standard self-training to explicitly address the distribution shift in domain adaptation. We show that our method provably works under the expansion assumption and demonstrate hard cases for feature adaptation and standard self-training. Self-training (or pseudo-labeling) is only one line of works in the semi-supervised learning literature. Future work can delve into the behaviors of other semi-supervised learning techniques including consistency regularization and data augmentation under distribution shift, and exploit them extensively for domain adaptation.

## Acknowledgements

This work was supported by the National Natural Science Foundation of China under Grants 62022050 and 62021002, Beijing Nova Program under Grant Z201100006820041, China's Ministry of Industry and Information Technology, the MOE Innovation Plan and the BNRist Innovation Fund.

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
