# A  Details in Section 4

## A.1  Proof of Theorem 1

In Section 4.1, we study CST theoretically. In Theorem 1, we show that when the population error of the target classifier $f_t$ on the source domain $P$ is low and $f_t$ is locally consistent, the source classifier $f_s$ is guaranteed to perform well on the target domain $Q$. We further show that the consistency (robustness) is guaranteed by the confidence of the model (Lemma 3). Finally, we show in Theorem 2 that the minimizer of an objective function consistent with the CST objective in Section 3.1 leads to small target loss of the source classifier $\mathrm{Err}_Q(f_s)$.

We first review the assumptions made in Section 4.1 in order to prove Theorem 1. Consider a $K$-way classification problem. $f : X \to [0,1]^K \in \mathcal{F}$ and $\tilde{f}(x) := \arg\max_i f(x)_{[i]}$. We first state the properties of source and target distributions $P$ and $Q$. Assume the source and target distributions are composed of $K$ sub-populations, each corresponding to one class, and the sub-populations of different classes have disjoint support. This indicates that a ground truth labeling function exists, which is a common assumption as in [7, 66, 11]. We also assume for simplicity of presentation that $P(y = i) = Q(y = i)$. Note that our techniques can be directly applied to the case where $\frac{P(y=i)}{Q(y=i)}$ is bounded as in [11].

**Assumption 1.** *Denote by $P_i$ and $Q_i$ the conditional distribution of $P$ and $Q$ given $y = i$. We assume that: (1) $P(y = i) = Q(y = i)$, and (2) the supports of $P_i$ and $P_j$ are disjoint for $i \neq j$.*

Our analysis relies on the *expansion assumption* [66, 11], which intuitively states that the data distribution has good continuity within each class. Therefore, the subset in the support of a class will connect to its neighborhood, enabling knowledge transfer between domains. Wei et al. [66] justifies this assumption on real-world datasets with BigGAN.

**Assumption 2** $((q, \epsilon)$-**constant expansion** [66]). *For any $x \in \mathcal{X}$, $\mathcal{N}(x)$ is defined as the neighboring set of $x$, $\mathcal{N}(x) = \{x' : d(x, x') \leq \xi\}$, where $d$ is a proper metric. $\mathcal{N}(A) := \cup_{x \in A}\mathcal{N}(x)$. We say $P$ and $Q$ satisfy $(q, \epsilon)$-constant expansion for some constants $q, \epsilon \in (0, 1)$, if for any set $A \in \mathcal{X}$ and any $i \in [K]$ with $\frac{1}{2} > P_{\frac{1}{2}(P_i+Q_i)}(A) > q$, we have $P_{\frac{1}{2}(P_i+Q_i)}(\mathcal{N}(A)\backslash A) > \min\{\epsilon, P_{\frac{1}{2}(P_i+Q_i)}(A)\}$.*

Based on this expansion assumption, we consider a *robustness-constrained* version of CST for now. In Theorem 2, we will show that the population robustness loss is closely related to the uncertainty of of CST. Denote by $f_s$ the source model and $f_t$ the model trained on the target with pseudo-labels. Let $R(f_t) := P_{\frac{1}{2}(P+Q)}(\{x : \exists x' \in \mathcal{N}(x), \tilde{f}_t(x) \neq \tilde{f}_t(x')\})$ represent the robustness [66] of $f_t$ on $P$ and $Q$. Suppose $\mathbb{E}_{(x,y)\sim Q}\mathbb{I}(\tilde{f}_s(x) \neq \tilde{f}_t(x)) \leq c$ and $R(f_t) \leq \rho$. Theorem 1 states that when $f_s$ and $f_t$ behave similarly on $Q$ ($f_t$ fits the pseudo-labels generated by $f_s$ on the target domain) and $f_t$ is robust to local changes in input, the minimizer of the cycle source error $\mathrm{Err}_P(f_t)$ will guarantee low error on the target domain $Q$.

**Theorem 1.** *Suppose Assumption 1 and Assumption 2 hold for $P$ and $Q$. For any $f_s$, $f_t$ satisfying $\mathbb{E}_{(x,y)\sim Q}\mathbb{I}(\tilde{f}_s(x) \neq \tilde{f}_t(x)) \leq c$ and $R(f_t) \leq \rho$, the expected error of $f_s$ on the target domain $Q$ is bounded,*

$$\mathrm{Err}_Q(f_s) \leq \mathrm{Err}_P(f_t) + c + 2q + \rho \; / \; \min\{\epsilon, q\}. \tag{17}$$

We now turn to the proof of Theorem 1. We want to show the error of $f_t$ on the source domain $P$ is close to the error of $f_s$ on the target domain $Q$. We first show that when the robustness error $R(f_t)$ is controlled, the error of $f_t$ on the source and the target will be close. This is done by analyzing the error on each sub-population $P_i$ and $Q_i$ separately. Then we use the fact that the losses of $f_s$ and $f_t$ are also close when their disagreement on the target domain is controlled to obtain the final result.

**Lemma 1** (Robustness on sub-populations). *Divide $[K]$ into $S_1$ and $S_2$, where for every $i \in S_1$, $\mathbb{E}_{(x,y)\sim\frac{1}{2}(P_i+Q_i)}\mathbb{I}(\exists x' \in \mathcal{N}(x), \tilde{f}_t(x) \neq \tilde{f}_t(x')) < \min\{\epsilon, q\}$, and for every $i \in S_2$, $\mathbb{E}_{(x,y)\sim\frac{1}{2}(P_i+Q_i)}\mathbb{I}(\exists x' \in \mathcal{N}(x), \tilde{f}_t(x) \neq \tilde{f}_t(x')) \geq \min\{\epsilon, q\}$. Under the condition of Theorem 1, we have*

$$\sum_{i \in S_1} P(y = i) \geq 1 - \frac{\rho}{\min\{\epsilon, q\}}. \tag{18}$$

*Proof of Lemma 1.* Suppose $\sum_{i \in S_1} P(y = i) < 1 - \frac{\rho}{\min\{\epsilon, q\}}$. Then we have $\sum_{i \in S_2} P(y = i) > \frac{\rho}{\min\{\epsilon, q\}}$, which implies

$$
\begin{aligned}
&\mathbb{E}_{(x,y) \sim \frac{1}{2}(P+Q)} \mathbb{I}(\exists x' \in \mathcal{N}(x), \tilde{f}_t(x) \neq \tilde{f}_t(x')) \\
&= \sum_{i \in [K]} \mathbb{E}_{(x,y) \sim \frac{1}{2}(P_i+Q_i)} \mathbb{I}(\exists x' \in \mathcal{N}(x), \tilde{f}_t(x) \neq \tilde{f}_t(x')) P(y = i) \\
&\geq \sum_{i \in S_2} \mathbb{E}_{(x,y) \sim \frac{1}{2}(P_i+Q_i)} \mathbb{I}(\exists x' \in \mathcal{N}(x), \tilde{f}_t(x) \neq \tilde{f}_t(x')) P(y = i) \\
&> \min\{\epsilon, q\} \sum_{i \in S_2} P(y = i) \\
&= \rho.
\end{aligned}
$$

Since we have $R(f_t) = \mathbb{E}_{(x,y) \sim \frac{1}{2}(P+Q)} \mathbb{I}(\exists x' \in \mathcal{N}(x), \tilde{f}_t(x) \neq \tilde{f}_t(x')) < \rho$, this forms a contradiction. $\qquad\square$

We have established that for a large proportion of the sub-populations, the robustness is guaranteed. The next lemma shows that for each sub-population where the robustness is guaranteed, $\mathrm{Err}_{P_i}(f_t)$ and $\mathrm{Err}_{Q_i}(f_t)$ is close to each other by invoking the expansion assumption [66].

**Lemma 2** (Accuracy propagates on robust sub-populations). *Under the condition of Theorem 1, if the sub-populations $P_i$ and $Q_i$ satisfy $\mathbb{E}_{(x,y) \sim \frac{1}{2}(P_i+Q_i)} \mathbb{I}(\exists x' \in \mathcal{N}(x), \tilde{f}_t(x) \neq \tilde{f}_t(x')) < \min\{\epsilon, q\}$, we have*

$$
|\mathrm{Err}_{P_i}(f_t) - \mathrm{Err}_{Q_i}(f_t)| \leq 2q. \tag{19}
$$

*Proof of Lemma 2.* We claim that either $\mathrm{Err}_{\frac{1}{2}(P_i+Q_i)}(f_t) \leq q$ or $\mathrm{Err}_{\frac{1}{2}(P_i+Q_i)}(f_t) \geq 1 - q$. On the one hand, if $\frac{1}{2} > \mathrm{Err}_{\frac{1}{2}(P_i+Q_i)}(f_t) > q$, by the $(q, \epsilon)$-expansion property (Definition 1), $P_{\frac{1}{2}(P_i+Q_i)}(\mathcal{N}(\{x : \tilde{f}_t(x) \neq i\}) \backslash \{x : \tilde{f}_t(x) \neq i\}) > \min\{\epsilon, q\}$. Note that in $\mathcal{N}(\{x : \tilde{f}_t(x) \neq i\}) \backslash \{x : \tilde{f}_t(x) \neq i\}$, $\tilde{f}_t(x) = i$. Thus, for $x$ in the set $\mathcal{N}(\{x : \tilde{f}_t(x) \neq i\}) \backslash \{x : \tilde{f}_t(x) \neq i\}$, there exists $x' \in \mathcal{N}(x)$, $\tilde{f}_t(x') \neq \tilde{f}_t(x) = i$.

$$
\begin{aligned}
R(f_t) &= \mathbb{E}_{(x,y) \sim \frac{1}{2}(P_i+Q_i)} \mathbb{I}(\exists x' \in \mathcal{N}(x), \tilde{f}_t(x) \neq \tilde{f}_t(x')) \\
&\geq \mathbb{E}_{(x,y) \sim \frac{1}{2}(P_i+Q_i)} \mathbb{I}(\exists x' \in \mathcal{N}(x), \tilde{f}_t(x) \neq \tilde{f}_t(x')) \mathbb{I}(x \in \mathcal{N}(\{x : \tilde{f}_t(x) \neq i\}) \backslash \{x : \tilde{f}_t(x) \neq i\}) \\
&= P_{\frac{1}{2}(P_i+Q_i)}(\mathcal{N}(\{x : \tilde{f}_t(x) \neq i\}) \backslash \{x : \tilde{f}_t(x) \neq i\}) \\
&> \min\{\epsilon, q\},
\end{aligned}
$$

which contradicts the condition that $R(f_t) < \min\{\epsilon, q\}$.

On the other hand, if $\frac{1}{2} \leq \mathrm{Err}_{\frac{1}{2}(P_i+Q_i)}(f_t) < 1 - q$, the argument is similar. By the $(q, \epsilon)$-expansion property (Definition 1), $P_{\frac{1}{2}(P_i+Q_i)}(\mathcal{N}(\{x : \tilde{f}_t(x) = i\}) \backslash \{x : \tilde{f}_t(x) = i\}) > \min\{\epsilon, q\}$. Note that in $\mathcal{N}(\{x : \tilde{f}_t(x) = i\}) \backslash \{x : \tilde{f}_t(x) = i\}$, $\tilde{f}_t(x) \neq i$. Thus, for $x$ in the set $\mathcal{N}(\{x : \tilde{f}_t(x) = i\}) \backslash \{x : \tilde{f}_t(x) = i\}$, there exists $x' \in \mathcal{N}(x)$, $i = \tilde{f}_t(x') \neq \tilde{f}_t(x)$.

$$
\begin{aligned}
R(f_t) &= \mathbb{E}_{(x,y) \sim \frac{1}{2}(P_i+Q_i)} \mathbb{I}(\exists x' \in \mathcal{N}(x), \tilde{f}_t(x) \neq \tilde{f}_t(x')) \\
&\geq \mathbb{E}_{(x,y) \sim \frac{1}{2}(P_i+Q_i)} \mathbb{I}(\exists x' \in \mathcal{N}(x), \tilde{f}_t(x) \neq \tilde{f}_t(x')) \mathbb{I}(x \in \mathcal{N}(\{x : \tilde{f}_t(x) = i\}) \backslash \{x : \tilde{f}_t(x) = i\}) \\
&= P_{\frac{1}{2}(P_i+Q_i)}(\mathcal{N}(\{x : \tilde{f}_t(x) = i\}) \backslash \{x : \tilde{f}_t(x) = i\}) \\
&> \min\{\epsilon, q\},
\end{aligned}
$$

which also contradicts the condition that $R(f_t) < \min\{\epsilon, q\}$.

Note that $\mathrm{Err}_{\frac{1}{2}(P_i+Q_i)}(f_t) = \frac{1}{2}\mathrm{Err}_{P_i}(f_t) + \frac{1}{2}\mathrm{Err}_{Q_i}(f_t)$. Also we have $\mathrm{Err}_{P_i}(f_t) \in [0, 1]$. In consequence, we have either $\mathrm{Err}_{P_i}(f_t), \mathrm{Err}_{Q_i}(f_t) \in [0, 2q]$ or $\mathrm{Err}_{P_i}(f_t), \mathrm{Err}_{Q_i}(f_t) \in [1 - 2q, 1]$, which completes the proof. $\qquad\square$

With Lemma 1 and Lemma 2 at hand, we can prove Theorem 1 by putting the analysis on each sub-population together.

*Proof of Theorem 1.*

$$\begin{aligned}
\mathrm{Err}_Q(f_t) &= \sum_{i \in [K]} \mathrm{Err}_{Q_i}(f_t) P(y = i) \\
&\leq \sum_{i \in [S_1]} \mathrm{Err}_{Q_i}(f_t) P(y = i) + \sum_{i \in [S_2]} P(y = i) \\
&\leq \sum_{i \in [S_1]} (\mathrm{Err}_{P_i}(f_t) + 2q) P(y = i) + \sum_{i \in [S_2]} P(y = i) \\
&\leq \mathrm{Err}_P(f_t) + 2q + \frac{\rho}{\min\{\epsilon, q\}},
\end{aligned}$$

where the second inequality holds due to Lemma 2, and the last holds due to Lemma 1. Also note that $\mathrm{Err}_Q(f_s) \leq \mathrm{Err}_Q(f_t) + \mathbb{E}_{(x,y)\sim Q}\mathbb{I}(\arg\max_{[i]} f_s(x)_{[i]} \neq \arg\max_{[i]} f_t(x)_{[i]})$ by the triangle inequality. Adding these two equations results in Theorem 1. □

## A.2 Proof of Theorem 2

To obtain finite-sample guarantee, we need additional assumptions on the function class $\mathcal{F}$.

**Assumption 3.** *The function class $\mathcal{F}$ satisfies the following properties: (1) $\mathcal{F}$ is closed to permutations of coordinates, (2) $0 \in \mathcal{F}$, and (3) each coordinate of $f$ is $L_f$-Lipschitz w.r.t. $d(\cdot, \cdot)$.*

This assumption is also standard since common models for multi-class classification are symmetric for each class. Setting all the weight parameters of neural networks to 0 will result in 0 output.

We review the definition of terms in Theorem 2. The ramp function $\psi_\gamma : \mathbb{R} \to [0,1]$ is defined as:

$$\psi_\gamma(x) = \begin{cases} 1, & x \leq 0 \\ 1 - \frac{x}{\gamma}, & 0 < x \leq \gamma \\ 0, & x > \gamma \end{cases} \tag{20}$$

The margin function is defined as $\mathcal{M}(v, y) = v_{[y]} - \max_{y' \neq y} v_{[y']}$ and $\mathcal{M}(v) = \max_y \{v_{[y]} - \max_{y' \neq y} v_{[y']}\}$. For multi-class classification problems, $\mathcal{M}(v)$ is closely related to the confidence, since it is equal to the difference between the largest and the second largest scores. The multi-class margin loss is composed of $\psi_\gamma(x)$ and $\mathcal{M}$: $l_\gamma(f(x), y) := \psi_\gamma(-\mathcal{M}(f(x), y))$. Denote by $L_{\widehat{P}, \gamma}(f_t)$ the empirical margin loss of $f_t$ on the source dataset $\widehat{P}$, $L_{\widehat{P}, \gamma}(f_t) = \mathbb{E}_{(x,y)\sim\widehat{P}} l_\gamma(f_t(x), y)$. To measure the inconsistency of $f_s$ and $f_t$, we extend the multi-class margin loss as $l_\gamma(f_s(x), f_t(x)) := \psi_\gamma(-\mathcal{M}(f_s(x), \tilde{f}_t(x)))$. Denote by $L_{\widehat{P}, \gamma}(f_t, f_s)$ the empirical margin inconsistency loss of $f_t$ and $f_s$ on the source dataset $\widehat{P}$, $L_{\widehat{P}, \gamma}(f_t, f_s) = \mathbb{E}_{(x,y)\sim\widehat{P}} l_\gamma(f_t(x), f_s(x))$.

Consider minimizing the following objective:

$$\min L_{\mathrm{CST}}(f_s, f_t) := \underbrace{L_{\widehat{P}, \gamma}(f_t)}_{\text{Cycle Loss}} + \underbrace{L_{\widehat{P}, \gamma}(f_t, f_s)}_{\text{Target Loss}} + \underbrace{1 - \mathbb{E}_{(x,y)\sim\frac{1}{2}(\widehat{P}+\widehat{Q})}\mathcal{M}(f_t(x))/\tau}_{\text{Uncertainty Loss}}. \tag{21}$$

Note that $L_{\widehat{P}, \gamma}(f_t)$ is the loss of $f_t$ on the source dataset (the cycle loss), and $L_{\widehat{P}, \gamma}(f_t, f_s)$ is the training error of $f_t$ on the target dataset. $\mathcal{M}(f_t(x))$ equals the difference between the largest and the second largest scores of $f_t(x)$, indicating the confidence of $f_t$. Thus, $1 - \mathbb{E}_{(x,y)\sim\frac{1}{2}(\widehat{P}+\widehat{Q})}\mathcal{M}(f_t(x))$ is the uncertainty of $f_t$ on the source and target datasets.

The following theorem shows that the minimizer of the training objective $L_{\mathrm{CST}}(f_s, f_t)$ guarantees low population error of $f_s$ on the target domain $Q$.

**Theorem 2.** *Under the condition of Theorem 1 and Assumption 3. For any solution of equation 13 and $\gamma > 0$, with probability larger than $1 - \delta$,*

$$\mathrm{Err}_Q(f_s) \leq L_{\mathrm{CST}}(f_s, f_t) + 2q + \frac{4K}{\gamma}\left[\widehat{\mathcal{R}}(\mathcal{F}|_{\widehat{P}}) + \widehat{\mathcal{R}}(\tilde{\mathcal{F}} \times \mathcal{F}|_{\widehat{Q}})\right] + \frac{2}{\tau}\left[\widehat{\mathcal{R}}(\mathcal{F}|_{\widehat{P}}) + \widehat{\mathcal{R}}(\mathcal{F}|_{\widehat{Q}})\right] + \zeta,$$

where $\zeta = O\left(\sqrt{\log(1/\delta)/n_s} + \sqrt{\log(1/\delta)/n_t}\right)$ is a low-order term. $\tilde{\mathcal{F}} \times \mathcal{F}$ refers to the function class $\{x \to f(x)_{[\tilde{f}'(x)]} : f, f' \in \mathcal{F}\}$. $\widehat{\mathcal{R}}(\mathcal{F}|_{\widehat{P}})$ denotes the empirical Rademacher complexity of function class $\mathcal{F}$ on dataset $\widehat{P}$.

We provide the function classes used in the proof. For a function class $f \in \mathcal{F} : \mathbb{R}^d \to [0,1]^K$, $\mathcal{F}_{[i]}$ denotes each coordinate of $\mathcal{F}$: $\mathcal{F}_{[i]} = \{x \to f(x)_{[i]} : f \in \mathcal{F}\}$. We also need other function classes based on $\mathcal{F}_{[i]}$. $\cup \mathcal{F}_{[i]}$ denotes the union of $\mathcal{F}_{[i]}$: $\cup \mathcal{F}_{[i]} = \cup_{i \in [K]} \mathcal{F}_{[i]}$. $\max_i \mathcal{F}_{[i]}$ is composed of the maximum coordinate of $f \in \mathcal{F}$ for all $x$: $\max_i \mathcal{F}_{[i]} = \{x \to \max_i f_{[i]}(x) : f \in \mathcal{F}\}$. $\max_{i' \neq \tilde{\mathcal{F}}} \mathcal{F}_{[i']}$ denotes the function class composed of the second largest coordinate of $f \in \mathcal{F}$ for all $x$: $\max_{i' \neq \tilde{\mathcal{F}}} \mathcal{F}_{[i']} = \{x \to \max_{i \neq \tilde{f}(x)} f_{[i]}(x) : f \in \mathcal{F}\}$, which we require to study the finite sample properties of the confidence loss $\mathcal{M}(f(x))$. $\tilde{\mathcal{F}} \times \mathcal{F}$ denotes the function class $\{x \to f_{\tilde{f}'(x)}(x) : f, f' \in \mathcal{F}\}$. The Rademacher complexity of $\mathcal{F}_{[i]}$ on set $S = \{x_j\}_{j=1}^n$ of size $n$ is: $\widehat{\mathcal{R}}(\mathcal{F}_{[i]}|_S) = \frac{1}{n}\mathbb{E}_{\sigma_j} \sup_{f \in \mathcal{F}} \sum_{j=1}^n \sigma_j f(x_j)_{[i]}$. The Rademacher complexity of $\cup \mathcal{F}_{[i]}$ is $\widehat{\mathcal{R}}(\cup \mathcal{F}_{[i]}|_S) = \frac{1}{n}\mathbb{E}_{\sigma_j} \sup_{f \in \mathcal{F}, i \in [K]} \sum_{j=1}^n \sigma_j f(x_j)_{[i]}$. The Rademacher complexity of $\max_i \mathcal{F}_{[i]}$ is $\widehat{\mathcal{R}}(\max_i \mathcal{F}_{[i]}|_S) = \frac{1}{n}\mathbb{E}_{\sigma_j} \sup_{f \in \mathcal{F}, i \in [K]} \sum_{j=1}^n \sigma_j \max_i f(x_j)_{[i]}$. We further denote by $\widehat{\mathcal{R}}(\mathcal{F}|_S)$ the sum of the Rademacher complexity of each $\mathcal{F}_{[i]}$, $\widehat{\mathcal{R}}(\mathcal{F}|_S) = \sum_{i=1}^K \widehat{\mathcal{R}}(\mathcal{F}_{[i]}|_S)$.

To prove Theorem 2, we first observe the relationship between the confidence objective $\mathbb{E}_{(x,y) \sim \frac{1}{2}(\widehat{P}+\widehat{Q})} \mathcal{M}(f_t(x))$ and the robustness constraint $R(f_t) := P_{\frac{1}{2}(P+Q)}(\{x : \exists x' \in \mathcal{N}(x), \max_{[i]} f_t(x) \neq \max_{[i]} f_t(x')\})$ in Theorem 1. In fact, as shown in Lemma 3, when the output of the model is confident on the source and target dataset, i.e. $\mathbb{E}_{(x,y) \sim \frac{1}{2}(\widehat{P}+\widehat{Q})} \mathcal{M}(f_t(x))$ is large, the model is also robust to the change in input.

**Lemma 3** (Confidence guarantees robustness). *Under the conditions of Theorem 2, we have*

$$R(f_t) \leq \frac{1 - \mathbb{E}_{(x,y) \sim \frac{1}{2}(P+Q)} \mathcal{M}(f_t(x))}{1 - 2L_f \xi}. \tag{22}$$

*Proof of Lemma 3.* We first note that when $\max_y\{f_t(x)_{[y]} - \max_{y' \neq y} f_t(x)_{[y']}\} > 2L_f \xi$, the $\arg\max_i f_t(x)_{[i]}$ will not change in the neighborhood $\mathcal{N}(x)$ since $f_{[i]}$ is $L_f$-Lipschitz for all $i$. Suppose $y^* = \arg\max_y f_t(x)_{[y]}$. For all $y' \neq y^*$ and $x' \in \mathcal{N}(x)$,

$$f_t(x')_{[y^*]} - f_t(x')_{[y']} > f_t(x)_{[y^*]} - L_f d(x, x') - (f_t(x')_{[y']} + L_f d(x, x')) \tag{23}$$

$$\geq \max_y\{f_t(x)_{[y]} - \max_{y' \neq y} f_t(x)_{[y']}\} - 2L_f d(x, x') \tag{24}$$

$$\geq 0. \tag{25}$$

Therefore, we have

$$R(f_t) \leq 1 - P_{\frac{1}{2}(P+Q)}\left(\mathcal{M}(f_t(x)) > 2L_f \xi\right) \tag{26}$$

$$\leq \frac{1 - \mathbb{E}_{(x,y) \sim \frac{1}{2}(P+Q)} \mathcal{M}(f_t(x))}{1 - 2L_f \xi}, \tag{27}$$

where the second inequality holds because $f_{[i]} \in [0,1]$, and $\mathcal{M}(f(x)) \in [0,1]$. $\square$

To obtain finite sample guarantee, we aim to show that each term in equation 13 is close to its population version. We first present Lemma 4, the classical result for multi-class classification.

**Lemma 4** (Lemma 3.1 of [41]). *Suppose $f \in \mathcal{F}$ and $\gamma > 0$, with probability at least $1 - \delta$ over the sampling of $\widehat{P}$, the following holds for all $f \in \mathcal{F}$ simultaneously,*

$$\text{Err}_P(f) \leq L_{\widehat{P},\gamma}(f) + \frac{4K}{\gamma}\widehat{\mathcal{R}}(\cup \mathcal{F}_{[i]}|_{\widehat{P}}) + O\left(\sqrt{\log(1/\delta)/n_s}\right). \tag{28}$$

We then extend Lemma 4 to study the finite sample properties of $L_{\widehat{Q},\gamma}(f_t, f_s)$ and $\mathbb{E}\mathcal{M}(f_t(x))$.

**Lemma 5.** *Suppose $f \in \mathcal{F}$ and $\gamma > 0$, with probability at least $1 - \delta$ over the sampling of $\widehat{P}$, the following holds for all $f \in \mathcal{F}$ simultaneously,*

$$\mathbb{E}_{(x,y)\sim P}\mathcal{M}(f(x)) \leq \mathbb{E}_{(x,y)\sim \widehat{P}}\mathcal{M}(f(x)) + 4\widehat{\mathcal{R}}(\mathcal{F}|_{\widehat{P}}) + O\left(\sqrt{\log(1/\delta) \ / \ n_s}\right). \qquad (29)$$

*Proof of Lemma 5.* By standard Rademacher complexity bound (Theorem 7 of Bartlett & Mendelson [5]), we have

$$\mathbb{E}_{(x,y)\sim P}\mathcal{M}(f(x)) \leq \mathbb{E}_{(x,y)\sim \widehat{P}}\mathcal{M}(f(x)) + 2\widehat{\mathcal{R}}(\mathcal{M} \circ \mathcal{F}|_{\widehat{P}}) + O\left(\sqrt{\log(1/\delta) \ / \ n_s}\right). \qquad (30)$$

Thus it remains to show $\widehat{\mathcal{R}}(\mathcal{M} \circ \mathcal{F}|_{\widehat{P}}) \leq 2\widehat{\mathcal{R}}(\mathcal{F}|_{\widehat{P}})$. In fact,

$$\begin{aligned}
\widehat{\mathcal{R}}(\mathcal{M} \circ \mathcal{F}|_{\widehat{P}}) &= \frac{1}{n_s}\mathbb{E}_\sigma \sup_{f\in\mathcal{F}} \sum_{i=1}^{n_s} \sigma_i \max_y\{f(x_i)_{[y]} - \max_{y'\neq y} f(x_i)_{[y']}\} \\
&= \frac{1}{n_s}\mathbb{E}_\sigma \sup_{f\in\mathcal{F}} \sum_{i=1}^{n_s} \sigma_i (\max_y f(x_i)_{[y]} - \max_{y'\neq \tilde{f}(x_i)} f(x_i)_{[y']}) \\
&\leq \frac{1}{n_s}\mathbb{E}_\sigma \sup_{f\in\mathcal{F}} \sum_{i=1}^{n_s} \sigma_i \max_y f(x_i)_{[y]} + \frac{1}{n_s}\mathbb{E}_\sigma \sup_{f\in\mathcal{F}} \sum_{i=1}^{n_s} \sigma_i \max_{y'\neq \tilde{f}(x_i)} f(x)_{[y']} \\
&= \widehat{\mathcal{R}}(\max_i \mathcal{F}_{[i]}|_{\widehat{P}}) + \widehat{\mathcal{R}}(\max_{i'\neq \tilde{\mathcal{F}}} \mathcal{F}_{[i']}|_{\widehat{P}}).
\end{aligned}$$

As will be shown in Lemma 7, both $\widehat{\mathcal{R}}(\max_i \mathcal{F}_{[i]}|_{\widehat{P}})$ and $\widehat{\mathcal{R}}(\max_{i'\neq \tilde{\mathcal{F}}} \mathcal{F}_{[i']}|_{\widehat{P}})$ are smaller than $\widehat{\mathcal{R}}(\mathcal{F}|_{\widehat{P}})$, which completes the proof. $\qquad \square$

**Lemma 6.** *Suppose $f_s, f_t \in \mathcal{F}$ and $\gamma > 0$, with probability at least $1 - \delta$ over the sampling of $\widehat{Q}$, the following holds for all $f_s, f_t \in \mathcal{F}$ simultaneously,*

$$\mathbb{E}_{(x,y)\sim Q}\mathbb{I}(f_t(x) \neq f_s(x)) \leq L_{\widehat{Q},\gamma}(f_t, f_s) + \frac{2K}{\gamma}\widehat{\mathcal{R}}(\tilde{\mathcal{F}} \times \mathcal{F}|_{\widehat{Q}}) + O\left(\sqrt{\log(1/\delta) \ / \ n_t}\right). \qquad (31)$$

*Proof of Lemma 6.* By the definition of multi-class margin loss, we have $\mathbb{E}_{(x,y)\sim Q}\mathbb{I}(f_t(x) \neq f_s(x)) \leq L_{Q,\gamma}(f_t, f_s)$. Denote by $\mathcal{G}$ the set of $\{x \to (-\mathcal{M}(f_t(x), f_s(x))) : f_t, f_s \in \mathcal{F}\}$. By standard Rademacher complexity bound, we have,

$$L_{Q,\gamma}(f_t, f_s) \leq L_{\widehat{Q},\gamma}(f_t, f_s) + 2\widehat{\mathcal{R}}(\psi_\gamma \circ \mathcal{G}|_{\widehat{Q}}) + O\left(\sqrt{\log(1/\delta) \ / \ n_t}\right).$$

By Talagrand contraction Lemma [59], $\widehat{\mathcal{R}}(\psi_\gamma \circ \mathcal{G}|_{\widehat{Q}}) \leq \frac{1}{\gamma}\widehat{\mathcal{R}}(\mathcal{G}|_{\widehat{Q}})$. Thus, it remains to show $\widehat{\mathcal{R}}(\mathcal{G}|_{\widehat{Q}}) \leq K\widehat{\mathcal{R}}(\tilde{\mathcal{F}} \times \mathcal{F}|_{\widehat{Q}})$. We have

$$\begin{aligned}
\widehat{\mathcal{R}}(\mathcal{G}|_{\widehat{Q}}) &= \frac{1}{n_t}\mathbb{E}_{\sigma_i} \sup_{f_s, f_t} \sum_{i=1}^{n_t} \sigma_i \mathcal{M}(f_t(x_i), \tilde{f}_s(x_i)) \\
&= \frac{1}{n_t}\mathbb{E}_{\sigma_i} \sup_{f_s, f_t} \sum_{i=1}^{n_t} \sigma_i \left( f_t(x_i)_{[\tilde{f}_s(x_i)]} - \max_{y'\neq \tilde{f}_s(x_i)} f_t(x_i)_{[y']} \right) \\
&\leq \frac{1}{n_t}\mathbb{E}_{\sigma_i} \sup_{f_s, f_t} \sum_{i=1}^{n_t} \sigma_i f_t(x_i)_{[\tilde{f}_s(x_i)]} + \frac{1}{n_t}\mathbb{E}_{\sigma_i} \sup_{f_s, f_t} \sum_{i=1}^{n_t} \sigma_i \max_{y'\neq \tilde{f}_s(x_i)} f_t(x_i)_{[y']} \\
&= \widehat{\mathcal{R}}(\tilde{\mathcal{F}} \times \mathcal{F}|_{\widehat{Q}}) + \frac{1}{n_t}\mathbb{E}_{\sigma_i} \sup_{f_s, f_t} \sum_{i=1}^{n_t} \sigma_i \max_{y'\neq \tilde{f}_s(x_i)} f_t(x_i)_{[y']}.
\end{aligned}$$

It remains to show $\frac{1}{n_t}\mathbb{E}_{\sigma_i} \sup_{f_s, f_t} \sum_{i=1}^{n_t} \sigma_i \max_{y'\neq \tilde{f}_s(x_i)} f_t(x_i)_{[y']} \leq (K-1)\widehat{\mathcal{R}}(\tilde{\mathcal{F}} \times \mathcal{F}|_{\widehat{Q}})$, which is done by noting the closure of $\mathcal{F}$ under the permutation of coordinates. Consider the permutation $\upsilon : \mathbb{R}^K \to \mathbb{R}^K : \upsilon(v)_{[i]} = v_{[i-1]}$ for $i \in [2, 3, \cdots K]$ and $\upsilon(v)_{[1]} = v_{[K]}$.

$$\frac{1}{n_t}\mathbb{E}_{\sigma_i} \sup_{f_s, f_t} \sum_{i=1}^{n_t} \sigma_i \max_{y'\neq \tilde{f}_s(x_i)} f_t(x_i)_{[y']} = \frac{1}{n_t}\mathbb{E}_{\sigma_i} \sup_{f_s, f_t} \sum_{i=1}^{n_t} \sigma_i \max_{k\in[K-1]} \upsilon^k f_t(x_i)_{[\tilde{f}_s(x_i)]}.$$

We have $\upsilon\mathcal{F} \subset \mathcal{F}$ by the closure of $\mathcal{F}$. Thus, $\tilde{\mathcal{F}} \times \upsilon\mathcal{F} \subset \tilde{\mathcal{F}} \times \mathcal{F}$. By Lemma 7, the Rademacher complexity of maximum of function classes is bounded with their sum, so we have $\frac{1}{n_t}\mathbb{E}_{\sigma_i}\sup_{f_s,f_t}\sum_{i=1}^{n_t}\sigma_i\max_{k\in[K-1]}\upsilon^k f_t(x_i)_{[\tilde{f}_s(x_i)]} \leq (K-1)\widehat{\mathcal{R}}(\tilde{\mathcal{F}}\times\mathcal{F}|_{\widehat{Q}})$. $\square$

The next lemma shows the relationship between function classes. We establish the Rademacher complexity bounds of $\cup\mathcal{F}_{[i]}$, $\max_i\mathcal{F}_{[i]}$, and $\max_{i'\neq\tilde{\mathcal{F}}}\mathcal{F}_{[i']}$. We show that the Rademacher complexity of these function classes can be bounded with $\widehat{\mathcal{R}}(\mathcal{F}|_S) = \sum_{i=1}^{K}\widehat{\mathcal{R}}(\mathcal{F}_{[i]}|_S)$.

**Lemma 7.** *Suppose* $\max_i\mathcal{F}_{[i]} = \{\max_i f_{[i]} : f \in \mathcal{F}\}$, $\cup\mathcal{F}_{[i]} = \{f_{[i]} : f \in \mathcal{F}, i \in [K]\}$, *and* $\max_{i'\neq\tilde{\mathcal{F}}}\mathcal{F}_{[i']} = \max_{i'\neq\tilde{\mathcal{F}}}\mathcal{F}_{[i']}|_{\widehat{P}} = \{x \to \max_{i\neq\tilde{f}(x)} f_{[i]}(x) : f \in \mathcal{F}\}$.

$$\widehat{\mathcal{R}}(\cup\mathcal{F}_{[i]}|_S) \leq \widehat{\mathcal{R}}(\mathcal{F}|_S), \ \widehat{\mathcal{R}}(\max_i\mathcal{F}_{[i]}|_S) \leq \widehat{\mathcal{R}}(\mathcal{F}|_S) \quad and \quad \widehat{\mathcal{R}}(\max_{i'\neq\tilde{\mathcal{F}}}\mathcal{F}_{[i']}|_S) \leq \widehat{\mathcal{R}}(\mathcal{F}|_S). \quad (32)$$

*Proof of Lemma 7.* Consider the $K = 2$ case. Then we can repeat the arguments for $K - 1$ times to get the final results.

For the first inequality, consider $\mathcal{F}'_{[i]} := \mathcal{F}_{[i]} \cup -\mathcal{F}_{[i]} = \{x \to \pm f(x) : f \in \mathcal{F}_{[i]}\}$. Then we have

$$\widehat{\mathcal{R}}(\mathcal{F}_{[1]}\cup\mathcal{F}_{[2]}|_S) = \frac{1}{n}\mathbb{E}_{\sigma_j}\sup_{f_{[i]}\in\mathcal{F}_{[1]}\cup\mathcal{F}_{[2]}}\sum_{j=1}^{n}\sigma_j f_{[i]}(x_j) \quad (33)$$

$$= \frac{1}{n}\mathbb{E}_{\sigma_j}\sup_{f'_{[i]}\in\mathcal{F}'_{[1]}\cup\mathcal{F}'_{[2]}}\sum_{j=1}^{n}\left|\sigma_j f'_{[i]}(x_j)\right| \quad (34)$$

$$\leq \frac{1}{n}\mathbb{E}_{\sigma_j}\sup_{f'_{[i]}\in\mathcal{F}'_{[1]}}\sum_{j=1}^{n}\left|\sigma_j f'_{[i]}(x_j)\right| + \frac{1}{n}\mathbb{E}_{\sigma_j}\sup_{f'_{[i]}\in\mathcal{F}'_{[2]}}\sum_{j=1}^{n}\left|\sigma_j f'_{[i]}(x_j)\right| \quad (35)$$

$$= \frac{1}{n}\mathbb{E}_{\sigma_j}\sup_{f_{[i]}\in\mathcal{F}_{[1]}}\sum_{j=1}^{n}\sigma_j f_{[i]}(x_j) + \frac{1}{n}\mathbb{E}_{\sigma_j}\sup_{f_{[i]}\in\mathcal{F}_{[2]}}\sum_{j=1}^{n}\sigma_j f_{[i]}(x_j) \quad (36)$$

$$= \widehat{\mathcal{R}}(\mathcal{F}_{[1]}|_S) + \widehat{\mathcal{R}}(\mathcal{F}_{[2]}|_S), \quad (37)$$

where equation equation 34 and equation 36 hold by the definition of $\mathcal{F}_{[i]}$.

For the second inequality, note that $\max\{x, y\} = \frac{x+y}{2} + \frac{|x-y|}{2}$. Then we apply Talagrand contraction lemma for the absolute value ($|\cdot|$ is 1-Lipschitz),

$$\widehat{\mathcal{R}}(\max_i\mathcal{F}_{[i]}|_S) = \frac{1}{n}\mathbb{E}_{\sigma_j}\sup_{f_{[1]}\in\mathcal{F}_{[1]}, f_{[2]}\in\mathcal{F}_{[2]}}\sum_{j=1}^{n}\sigma_j\left(\frac{f_{[1]}(x_j)+f_{[2]}(x_j)}{2} + \frac{|f_{[1]}(x_j)-f_{[2]}(x_j)|}{2}\right)$$

$$\leq \frac{1}{2}\widehat{\mathcal{R}}(\mathcal{F}_{[1]}|_S) + \frac{1}{2}\widehat{\mathcal{R}}(\mathcal{F}_{[2]}|_S) + \frac{1}{2}\widehat{\mathcal{R}}(|\mathcal{F}_{[1]}-\mathcal{F}_{[2]}||_S)$$

$$\leq \frac{1}{2}\widehat{\mathcal{R}}(\mathcal{F}_{[1]}|_S) + \frac{1}{2}\widehat{\mathcal{R}}(\mathcal{F}_{[2]}|_S) + \frac{1}{2}\widehat{\mathcal{R}}(\mathcal{F}_{[1]}-\mathcal{F}_{[2]}|_S)$$

$$\leq \widehat{\mathcal{R}}(\mathcal{F}_{[1]}|_S) + \widehat{\mathcal{R}}(\mathcal{F}_{[2]}|_S).$$

For the third inequality, observe that the second largest of the set $\{x, y, z\}$ can be expressed as $\max\{\min\{x, y\}, \min\{\max\{x, y\}, z\}\}$. Following argument similar to the second inequality gives the proof. $\square$

Now equipped with the lemmas above, we are ready to prove Theorem 2.

*Proof of Theorem 2.* By Lemmas 4, 5, and 6, we have the following inequalities hold with probability larger than $1 - \frac{\delta}{3}$,

$$\text{Err}_P(f) \leq L_{\widehat{P},\gamma}(f) + \frac{4K}{\gamma}\widehat{\mathcal{R}}(\cup\mathcal{F}_{[i]}|_{\widehat{P}}) + O\left(\sqrt{\log(1/\delta)/n_s}\right). \quad (38)$$

$$\mathbb{E}_{(x,y)\sim\frac{1}{2}(P+Q)}\mathcal{M}(f(x)) \leq \mathbb{E}_{(x,y)\sim\frac{1}{2}(\widehat{P}+\widehat{Q})}\mathcal{M}(f(x)) + 2\widehat{\mathcal{R}}(\mathcal{F}|_{\widehat{P}}) + 2\widehat{\mathcal{R}}(\mathcal{F}|_{\widehat{Q}}) \tag{39}$$
$$+ O\left(\sqrt{\log(1/\delta) / n_s} + \sqrt{\log(1/\delta) / n_t}\right).$$

$$\mathbb{E}_{(x,y)\sim Q}\mathbb{I}(f_t(x) \neq f_s(x)) \leq L_{\widehat{Q},\gamma}(f_t, f_s) + \frac{2K}{\gamma}\widehat{\mathcal{R}}(\tilde{\mathcal{F}} \times \mathcal{F}|_{\widehat{Q}}) + O\left(\sqrt{\log(1/\delta) / n_t}\right). \tag{40}$$

We also have the following due to Lemma 3,

$$R(f_t) \leq \frac{1 - \mathbb{E}_{(x,y)\sim\frac{1}{2}(P+Q)}\mathcal{M}(f_t(x))}{1 - 2L_f\xi}. \tag{41}$$

We use equation 39 and equation 40 as conditions. Plugging equation 38, equation 39, equation 40, and equation 41 into Theorem 1 and applying a union bound complete the proof of Theorem 2. □

## A.3  Details in Section 4.2

We instantiate the domain adaptation setting in a quadratic neural network that allows us to compare various properties of the related algorithms. For a specific data distribution, we prove that (1) cycle self-training recovers target ground truth, and (2) both feature adaptation and standard self-training fail on the same distribution.

### A.3.1  Setup

We study a quadratic neural network composed of a feature extractor $\phi \in \mathbb{R}^{d\times m}$ and a head $\theta \in \mathbb{R}^m$. $f_{\theta,\phi}(x) = g_\theta(h_\phi(x))$, where $g_\theta(z) = \theta^\top z$ and $h_\phi(x) = (\phi^\top x) \odot (\phi^\top x)$, $\odot$ is element-wise product. In training, we use the squared loss $\ell(f(x), y) = (f(x) - y)^2$. In testing, we map $f(x)$ to the nearest point in the output space: $\tilde{f}(x) := \arg\min_{y\in\{-1,0,1\}} |y - f(x)|$. Denote the expected error by $\mathrm{Err}_Q(\theta, \phi) := \mathbb{E}_{(x,y)\sim Q}\mathbb{I}(\tilde{f}_{\theta,\phi}(x) \neq y)$.

**Structural Covariate Shift and Label Shift.** In domain adaptation, the source domain can have *multiple solutions* but we aim to learn the solution which works on the target domain [34]. Recent works also pointed out the source and the target label distributions are often different in real-world applications [74]. Following these properties, we design the underlying distributions $p$ and $q$ as shown in Table 6 to allow both structural covariate shift and label shift.

Table 6: Comparison of the design of the source and target.

| Distribution | $-1$ | $+1$ | $0$ |
|---|---|---|---|
| Source $p$ | 0.05 | 0.05 | 0.90 |
| Target $q$ | 0.25 | 0.25 | 0.50 |

We study the following source distribution $P$. $x_{[1]}$ and $x_{[2]}$ are sampled *i.i.d.* from distribution $p$, and for $i \in [3, d]$, $x_{[i]} = \sigma_i \times x_{[2]}$. $\sigma_i \in \{\pm 1\}$ uniformly. In the target domain, $x_{[1]}$ and $x_{[2]}$ are sampled *i.i.d.* from distribution $q$, and for $i \in [3, d]$, $x_{[i]} = \sigma_i \times x_{[1]}$. $\sigma_i \in \{\pm 1\}$ uniformly. We also assume realizability: $y = x_{[1]}^2 - x_{[2]}^2$ for both source and target. For simplicity, we assume access to infinite *i.i.d.* examples of $P$ ($n_s = \infty$) and $n_t$ *i.i.d.* examples of $Q$. Therefore, the empirical loss and the population loss on the source domain are the same $L_P = L_{\widehat{P}}$.

Note that since $x_{[i]}^2 = x_{[2]}^2$ for all $i \in [3, d]$ in the source domain, $y = x_{[1]}^2 - x_{[i]}^2$ for all $i \in [2, d]$ are solutions to the source domain but only $y = x_{[1]}^2 - x_{[2]}^2$ works on the target domain. We visualize the setting when $d = 3$ in Figure 4.

### A.3.2  Algorithms

We compare the baseline algorithms (feature adaptation and self-training) in Section 2 with the proposed CST. We study the norm-constrained versions of these algorithms.

**Feature Adaptation** chooses the source solution minimizing the distance between source and target feature distributions. We use total variation (TV) distance [7]: $d_{\text{TV}}(h_\sharp \widehat{P}, h_\sharp \widehat{Q}) = \sup_{E \subset \mathcal{Z}} |h_\sharp \widehat{P}(E) - h_\sharp \widehat{Q}(E)|$.

$$\hat{\theta}_{\text{FA}}, \hat{\phi}_{\text{FA}} = \arg\min_{\hat{\theta}_s, \hat{\phi}_s} d_{\text{TV}}(h_\sharp \widehat{P}, h_\sharp \widehat{Q}), \tag{42}$$

$$\text{s.t. } \hat{\theta}_s, \hat{\phi}_s = \arg\min_{\theta, \phi} \|\theta\|_2^2 + \|\phi\|_F^2, \text{ s.t. } L_P(\theta, \phi) = 0.$$

**Standard Self-Training** first trains a source model,

$$\hat{\theta}_s, \hat{\phi}_s = \arg\min_{\theta, \phi} \|\theta\|_2^2 + \|\phi\|_F^2, \text{ s.t. } L_P(\theta, \phi) = 0. \tag{43}$$

Then it trains the model on the source and target datasets jointly with source ground-truths and target pseudo-labels,

$$\hat{\theta}_{\text{ST}}, \hat{\phi}_{\text{ST}} = \arg\min_{\theta, \phi} \|\theta\|_2^2 + \|\phi\|_F^2, \tag{44}$$

$$\text{s.t. } L_P(\theta, \phi) + \mathbb{E}_{x \sim \widehat{Q}} \ell(f_{\theta, \phi}(x), f_{\hat{\theta}_s, \hat{\phi}_s}(x)) = 0.$$

**Cycle Self-Training.** Following Section 3.1, we train the source head $\theta_s$, and then train another head $\hat{\theta}_t(\phi)$ on the target dataset $\widehat{Q}$ with pseudo-labels generated by $\theta_s$:

$$\hat{\theta}_t(\phi) = \arg\min_{\theta} \|\theta\|_2^2, \text{ s.t. } \mathbb{E}_{x \in \widehat{Q}} \ell(f_{\theta, \phi}(x), f_{\theta_s, \phi}(x)) = 0.$$

Finally we update the feature extractor $\phi$ to enforce consistent predictions of $\hat{\theta}_t(\phi)$ and $\theta_s$ on the source dataset:

$$\hat{\theta}_{\text{CST}}, \hat{\phi}_{\text{CST}} = \arg\min_{\theta_s, \phi} \|\theta_s\|_2^2 + \|\phi\|_F^2, \tag{45}$$

$$\text{s.t. } L_P(\theta_s, \phi) + \mathbb{E}_{x \in P} \ell(g_{\theta_s}(h_\phi(x)), g_{\hat{\theta}_t(\phi)}(h_\phi(x))) = 0.$$

The following theorems show that both feature adaptation and standard self-training fail. The intuition is that the *ideal* solution that works on both source and target $y = x_{[1]}^2 - x_{[2]}^2$ has larger distance $d_{\text{TV}}$ in the feature space than other solutions $y = x_{[1]}^2 - x_{[i]}^2$, so feature adaptation will not prefer the ideal solution. Standard self-training also fails because it will choose randomly among $y = x_{[1]}^2 - x_{[i]}^2$.

**Theorem 3.** *For any $\epsilon \in (0, 0.5)$, the following statements are true for feature adaptation and standard self-training:*

- *For any failure rate $\xi > 0$, and target dataset of size $n_t > \Theta(\log \frac{1}{\xi})$, with probability at least $1 - \xi$ over the sampling of target data, the source solution $\hat{\theta}_{\text{FA}}, \hat{\phi}_{\text{FA}}$ found by feature adaptation fails on the target domain:*

$$\text{Err}_Q(\hat{\theta}_{\text{FA}}, \hat{\phi}_{\text{FA}}) \geq \epsilon. \tag{46}$$

- *With probability at least $1 - \frac{1}{d-1}$ over the training the source solution, the solution $(\hat{\theta}_{\text{ST}}, \hat{\phi}_{\text{ST}})$ of standard self-training satisfies*

$$\text{Err}_Q(\hat{\theta}_{\text{ST}}, \hat{\phi}_{\text{ST}}) \geq \epsilon. \tag{47}$$

In comparison, we show that CST can recover the ground truth with high probability.

**Theorem 4.** *For any failure rate $\xi > 0$, and target dataset of size $n_t > \Theta(\log \frac{1}{\xi})$, with probability at least $1 - \xi$ over the sampling of target data, the feature extractor $\hat{\phi}_{\text{CST}}$ found by CST and the head $\hat{\theta}_{\text{CST}}$ recovers the ground truth of the target dataset:*

$$\text{Err}_Q(\hat{\theta}_{\text{CST}}, \hat{\phi}_{\text{CST}}) = 0. \tag{48}$$

Intuitively, CST successfully learns the *transferable feature* $x_{[1]}^2 - x_{[2]}^2$ because it enforces the generalization of the head $\hat{\theta}_t(\phi)$ on the source data.

### A.4 Proof of Theorem 3

We first describe the insights of the proof. As shown in Lemma 8, every source solution can be categorized into $d - 1$ classes according to the coordinate $l$ of the learned weight $\phi$. Among those $d - 1$ classes, only $l = 2$ works on the target domain and $l \in \{3, \cdots d\}$ do not work on the target. We then show that in feature adaptation, $l \in \{3, \cdots d\}$ results in smaller distance between source and target feature distributions as a result of Lemma 9, thus feature adaptation will choose $l \in \{3, \cdots d\}$. On the other hand, standard self-training with randomly select $l$ in the possible $d - 1$ choices, but only $l = 2$ works.

**Lemma 8.** *Under the condition of Section 4, any solution $\theta, \phi$ to the Source Only problem*

$$\min_{\theta, \phi} \|\theta\|_2^2 + \|\phi\|_F^2, \text{ s.t. } L_P(\theta, \phi) = 0. \tag{49}$$

*must have the following form:* $\exists i, j \in \{2, 3, \cdots m\}$, $l \in \{2, 3, \cdots d\}$, $\phi_i = 2^{\frac{1}{6}} e_1$, $\phi_j = 2^{\frac{1}{6}} e_l$, $\phi_k = 0$ *for* $k \neq i, j$, *and* $\theta_i = \theta_j = 2^{-\frac{1}{3}}$, $\theta_k = 0$ *for* $k \neq i, j$.

*Proof of Lemma 8.* Define the symmetric matrix $A = \sum_{i=1}^{m} \theta_i \phi_i \phi_i^{\top}$, then the networks can be represented by $A$: $f_{\theta, \phi}(x) = x^{\top} A x$. We show that $A_{ij} = 0$ for $i \neq j$ if $f_{\theta, \phi}$ recovers source ground truth.

First, for $i, j > 1$, let $x_1 = \mathbf{1}$, $x_2 = \mathbf{1} - 2e_i - 2e_j$, $x_3 = \mathbf{1} - 2e_i$, and $x_4 = \mathbf{1} - 2e_j$, where $\{e_i\}$ are the standard bases. Since the source ground truth $y = x_{[1]}^2 - x_{[2]}^2$, and $x_{[k]} = \pm 1 x_{[2]}$ for $k \in \{3, 4, \cdots d\}$, $y_1 = y_2 = y_3 = y_4$.

$$y_1 + y_2 + y_3 - y_4 = x_1^{\top} A x_1 + x_2^{\top} A x_2 - x_3^{\top} A x_3 - x_4^{\top} A x_4 \tag{50}$$

$$= 2\mathbf{1}^{\top} A \mathbf{1} + 4A_{ii} + 4A_{jj} - 4\mathbf{1}^{\top} A e_i - 4\mathbf{1}^{\top} A e_j + 8A_{ij} \tag{51}$$

$$- (\mathbf{1}^{\top} A \mathbf{1} + 4A_{ii} + 4A_{jj} - 4\mathbf{1}^{\top} A e_i - 4\mathbf{1}^{\top} A e_j) \tag{52}$$

$$= 8A_{ij} = 0. \tag{53}$$

We then show $A_{1,j} = 0$ for $j \in \{2, 3, \cdots d\}$ using the fact that $y_1 = y_4$.

$$y_1 - y_4 = x_1^{\top} A x_1 - x_4^{\top} A x_4 \tag{54}$$

$$= \mathbf{1}^{\top} A \mathbf{1} - (\mathbf{1}^{\top} A \mathbf{1} - 4\mathbf{1}^{\top} A e_j + 4A_{jj}) \tag{55}$$

$$= 4\mathbf{1}^{\top} A e_j - 4A_{jj} \tag{56}$$

$$= 0. \tag{57}$$

From equation 53 we know $A_{ij} = 0$ if $i \neq 1$. Then we also have $A_{1j} = A_{j1} = 0$. Therefore we can write $y$ in the following form: $y = x^{\top} A x = \sum_{i=1}^{d} A_{ii} x_{[i]}^2$ We also have the source ground truth $y = x_{[1]}^2 - x_{[2]}^2$, and $x_{[k]} = \pm 1 x_{[2]}$ for $k \in \{3, 4, \cdots d\}$. Then $A$ must satisfy $A_{11} = 1$, $\sum_{i=2}^{d} A_{ii} = -1$, and all other entries of $A$ equals to 0.

We have found the form of source ground truth matrix $A$. It suffices to show that the minimal norm solution of $\theta$ and $\phi$ subject to the form of $A$ must be in the form of Lemma 8.

$$\|\theta\|_2^2 + \|\phi\|_F^2 = \sum_{i}^{m} \theta_i^2 + \frac{1}{2}\|\phi_i\|_2^2 + \frac{1}{2}\|\phi_i\|_2^2 \tag{58}$$

$$\geq \sum_{i}^{m} 3 \cdot 2^{\frac{2}{3}} \left(|\theta_i| \|\phi_i\|_2^2\right)^{\frac{2}{3}} \tag{59}$$

$$\geq 3 \cdot 2^{\frac{2}{3}} \left(\sum_{i:\theta_i > 0} \theta_i \|\phi_i\|_2^2\right)^{\frac{2}{3}} + 3 \cdot 2^{\frac{2}{3}} \left(\sum_{i:\theta_i \leq 0} -\theta_i \|\phi_i\|_2^2\right)^{\frac{2}{3}} \tag{60}$$

$$= 3 \cdot 2^{\frac{2}{3}} \left(\sum_{i:A_{ii} > 0} A_{ii}\right)^{\frac{2}{3}} + 3 \cdot 2^{\frac{2}{3}} \left(\sum_{i:A_{ii} \leq 0} -A_{ii}\right)^{\frac{2}{3}} = 3 \cdot 2^{\frac{5}{3}}. \tag{61}$$

The first inequality holds due to AM-GM inequality, where it takes equality iff $\theta_i^2 = \frac{1}{2}\|\phi_i\|_2^2$ for all $i$. The second inequality holds due to Jensen inequality. The situation where both inequality take equality is exactly the form of Lemma 8. $\square$

**Lemma 9.** *Suppose $\widehat{Q} = \{x_i^t\}_{i=1}^{n_t}$ are i.i.d. samples from target distribution Q, then with high probability, $\mathbb{E}_{(x,y)\sim\widehat{Q}}\mathbb{I}(x_{[l]} = 0)$ is close to 0.5:*

$$P\left(\left|\mathbb{E}_{(x,y)\sim\widehat{Q}}\mathbb{I}(x_{[l]} = 0) - 0.5\right| > t\right) \le e^{-2n_t t^2} \tag{62}$$

*Proof of Lemma 9.* Since each coordinate of $x$ follows $q$, $\mathbb{I}(x_{[l]} = 0) - 0.5$ is a sub-Gaussian variable with $\sigma = 0.5$. We then apply standard Hoeffding's inequality to complete the proof. $\square$

*Proof of Theorem 3.* We have the conclusion of Lemma 8. For simplicity we suppose without loss of generality that the source solution has the following form: $\exists\, l \in \{2, 3, \cdots d\}$, $\phi_1 = 2^{\frac{1}{6}}e_1$, $\phi_2 = 2^{\frac{1}{6}}e_l$, $\phi_k = 0$ for $k \in \{3, \cdots m\}$, and $\theta_1 = \theta_2 = 2^{-\frac{1}{3}}$, $\theta_k = 0$ for $k \in \{3, \cdots m\}$. Then these solutions can be categorized into two classes: (1) When $l = 2$, the source solution also works on the target, i.e. $L_Q(\theta, \phi) = 0$. (2) When $l \in \{3, \cdots d\}$, the source solution does not work on the target,

$$\text{Err}_Q(\theta, \phi) = 1 - \mathbb{E}_{(x,y)\sim Q}\mathbb{I}(f_{\theta,\phi}(x) = y) \tag{63}$$

$$= 1 - \mathbb{E}_{(x,y)\sim Q}\mathbb{I}(x_{[l]}^2 = x_{[2]}^2) \tag{64}$$

$$= 1 - \mathbb{E}_{(x,y)\sim Q}\mathbb{I}(x_{[l]} = 0 \text{ and } x_{[2]} = 0) - \mathbb{E}_{(x,y)\sim Q}\mathbb{I}(x_{[l]} \ne 0 \text{ and } x_{[2]} \ne 0) \tag{65}$$

$$= 0.5 \tag{66}$$

To prove that feature adaptation learns the solution that does not work on the target domain, we show that with high probability, the solution belonging to situation (1) has larger total variation between source and target feature distributions $h_\sharp P$ and $h_\sharp \widehat{Q}$. In fact, the distributions of $h_\sharp P$ are the same for solutions in situation (1) and situation (2):

$$\mathbb{E}_{(x,y)\sim P}\left(h_\phi(x) = (z_1, z_2)\right) = \begin{cases} 0.81, & (z_1, z_2) = 2^{\frac{1}{3}}(0, 0) \\ 0.09, & (z_1, z_2) = 2^{\frac{1}{3}}(0, 1) \\ 0.09, & (z_1, z_2) = 2^{\frac{1}{3}}(1, 0) \\ 0.01, & (z_1, z_2) = 2^{\frac{1}{3}}(1, 1) \end{cases}$$

For the target dataset, the distribution of features is different for solutions from situation (1) and situation (2). When $l = 2$, denote by $h_{1\sharp}\widehat{Q}$ the feature distribution.

$$\mathbb{E}_{(x,y)\sim\widehat{Q}}\left(h_\phi(x) = (z_1, z_2)\right) = \begin{cases} \mathbb{E}_{(x,y)\sim\widehat{Q}}^2\mathbb{I}(x_{[l]} = 0), & (z_1, z_2) = 2^{\frac{1}{3}}(0, 0) \\ \mathbb{E}_{(x,y)\sim\widehat{Q}}\mathbb{I}(x_{[l]} = 0)\mathbb{E}_{(x,y)\sim\widehat{Q}}\mathbb{I}(x_{[l]} \ne 0), & (z_1, z_2) = 2^{\frac{1}{3}}(0, 1) \\ \mathbb{E}_{(x,y)\sim\widehat{Q}}\mathbb{I}(x_{[l]} = 0)\mathbb{E}_{(x,y)\sim\widehat{Q}}\mathbb{I}(x_{[l]} \ne 0), & (z_1, z_2) = 2^{\frac{1}{3}}(1, 0) \\ \mathbb{E}_{(x,y)\sim\widehat{Q}}^2\mathbb{I}(x_{[l]} \ne 0), & (z_1, z_2) = 2^{\frac{1}{3}}(1, 1) \end{cases}$$

When $l \in \{3, \cdots d\}$, denote by $h_{2\sharp}\widehat{Q}$ the feature distribution. since $x_{[l]} = x_{[1]}$ in the target domain, $(z_1, z_2)$ can only be $2^{\frac{1}{3}}(0, 0)$ or $2^{\frac{1}{3}}(1, 1)$,

$$\mathbb{E}_{(x,y)\sim\widehat{Q}}\left(h_\phi(x) = (z_1, z_2)\right) = \begin{cases} \mathbb{E}_{(x,y)\sim\widehat{Q}}\mathbb{I}(x_{[l]} = 0), & (z_1, z_2) = 2^{\frac{1}{3}}(0, 0) \\ \mathbb{E}_{(x,y)\sim\widehat{Q}}\mathbb{I}(x_{[l]} \ne 0), & (z_1, z_2) = 2^{\frac{1}{3}}(1, 1) \end{cases}$$

We then instantiate Lemma 9 with $t = 0.14$: With probability at least $1 - \delta$, $0.36 < \mathbb{E}_{(x,y)\sim\widehat{Q}}\mathbb{I}(x_{[l]} = 0) < 0.64$ for any $n_t \ge C\log\left(\frac{1}{\delta}\right)$, where $C > 26$ is a constant. Finally, we show that $d_{\text{TV}}(h_\sharp P, h_{2\sharp}\widehat{Q}) < d_{\text{TV}}(h_\sharp P, h_{1\sharp}\widehat{Q})$ as long as $0.36 < \mathbb{E}_{(x,y)\sim\widehat{Q}}\mathbb{I}(x_{[l]} = 0) < 0.64$ to prove that

feature adaptation will select solutions in situation (2).

$$d_{\text{TV}}(h_\sharp P, h_{1\sharp}\widehat{Q}) = \frac{1}{2}\left|\mathbb{E}^2_{(x,y)\sim\widehat{Q}}\mathbb{I}(x_{[l]}=0)-0.81\right| + \frac{1}{2}\left|\mathbb{E}^2_{(x,y)\sim\widehat{Q}}\mathbb{I}(x_{[l]}\neq 0)-0.01\right| \tag{67}$$

$$+ \left|\mathbb{E}_{(x,y)\sim\widehat{Q}}\mathbb{I}(x_{[l]}=0)\mathbb{E}_{(x,y)\sim\widehat{Q}}\mathbb{I}(x_{[l]}\neq 0)-0.09\right| \tag{68}$$

$$=0.81 - \mathbb{E}^2_{(x,y)\sim\widehat{Q}}\mathbb{I}(x_{[l]}=0) \tag{69}$$

$$>0.9 - \mathbb{E}_{(x,y)\sim\widehat{Q}}\mathbb{I}(x_{[l]}=0) \tag{70}$$

$$=\frac{1}{2}\left|\mathbb{E}_{(x,y)\sim\widehat{Q}}\mathbb{I}(x_{[l]}=0)-0.81\right| + \frac{1}{2}\left|\mathbb{E}_{(x,y)\sim\widehat{Q}}\mathbb{I}(x_{[l]}\neq 0)-0.01\right| \tag{71}$$

$$=d_{\text{TV}}(h_\sharp P, h_{2\sharp}\widehat{Q}), \tag{72}$$

when $0.36 < \mathbb{E}_{(x,y)\sim\widehat{Q}}\mathbb{I}(x_{[l]}=0) < 0.64$, which completes the proof of feature adaptation.

In standard self-training, when training the source solution, the probability of $l$ equalling each value in $\{2,3,\cdots d\}$ is the same, but only $l=2$ is the solution working on the source domain. Then when training on the source ground truth and target pseudo-labels, the model will make $l$ unchanged. Thus the probability of recovering the target ground truth is only $\frac{1}{d-1}$. $\square$

### A.5 Proof of Theorem 4

Similar to the proof of Theorem 3, we use the conclusion of Lemma 8 to show that $l=2$ indicates the source solution that works on the target domain, while the solutions corresponding to $l \in \{3,\cdots d\}$ will have large error on the target domain. Then we show that only $l=2$ makes the training objective of cycle self-training $L_{\text{CST}}=0$. This is due to the fact that $l=2$ will make the spans of source and target features identical, while $l \in \{3,\cdots d\}$ makes the spans of source and target features different and thus $\hat{\theta}_t(\phi) \neq \theta_s$.

*Proof of Theorem 4.* We still use Lemma 8. To prove that cycle self-training recovers the target ground truth, it suffices to show that $\mathbb{E}_{x\in P}\ell(g_{\theta_s}(h_\phi(x)), g_{\hat{\theta}_t(\phi)}(h_\phi(x))) = 0$ when $l=2$, and $\mathbb{E}_{x\in P}\ell(g_{\theta_s}(h_\phi(x)), g_{\hat{\theta}_t(\phi)}(h_\phi(x))) \neq 0$ when $l \in \{3,\cdots d\}$.

When $l \in \{3,\cdots d\}$, since $x^2_{[1]} = x^2_{[l]}$ in the target domain, the target pseudo-labels are all 0. Then we solve the problem $\hat{\theta}_t(\phi) = \arg\min_\theta \|\theta\|^2_2$, s.t. $\mathbb{E}_{x\in\widehat{Q}}\ell(f_{\theta_s,\phi}(x), f_{\theta,\phi}(x))$ to get the target classifier $\hat{\theta}_t(\phi)$. Since we want the target solution with minimal norm, $\hat{\theta}_t(\phi) = 0$, and we can calculate $L_{\text{CST}}$ as follows:

$$\mathbb{E}_{x\in P}\ell(g_{\theta_s}(h_\phi(x)), g_{\hat{\theta}_t(\phi)}(h_\phi(x))) = \mathbb{E}_{(x,y)\sim P}(y - f_{\hat{\theta}_t(\phi),\phi}(x))^2 = \mathbb{E}_{(x,y)\sim P}y^2 = 0.18. \tag{73}$$

When $l=2$, we show that $2^{\frac{1}{3}}e_1 + 2^{\frac{1}{3}}e_2, 2^{\frac{1}{3}}e_1, 2^{\frac{1}{3}}e_2$ and 0 all appear in the target feature set with high probability. The probability that the target feature set does not contain each one in $2^{\frac{1}{3}}e_1 + 2^{\frac{1}{3}}e_2, 2^{\frac{1}{3}}e_1, 2^{\frac{1}{3}}e_2$ and 0 equals to $\left(\frac{3}{4}\right)^{n_t}$. Therefore with a union bound we can show that $2^{\frac{1}{3}}e_1 + 2^{\frac{1}{3}}e_2, 2^{\frac{1}{3}}e_1, 2^{\frac{1}{3}}e_2$ and 0 all appear in the target feature set with probability at least $1 - 4\left(\frac{3}{4}\right)^{n_t}$. In this case, $\hat{\theta}_t(\phi) = \arg\min_\theta \|\theta\|^2_2$, s.t. $\mathbb{E}_{x\in\widehat{Q}}\ell(f_{\theta_s,\phi}(x), f_{\theta,\phi}(x))$ results in $\hat{\theta}_t(\phi) = \theta_s$, which means if $n_t > \Theta(\log\frac{1}{\xi})$, with probability at least $1 - \xi$ over the sampling of target data,

$$L_{\text{CST}} = \mathbb{E}_{x\in P}\ell(g_{\theta_s}(h_\phi(x)), g_{\hat{\theta}_t(\phi)}(h_\phi(x))) = 0. \tag{74}$$

$\square$

# B  Implementation Details

We use PyTorch [44] and run each experiment with 2080Ti GPUs. CBST, KLD, and IA results are from their original papers. We use the highest results in the literature for DANN, MCD, CDAN, and MDD. VAT, FixMatch, MixMatch and DIRT-T are adapted to our datasets from the official code. We adopt the pre-trained ResNet models provided in torchvision. For BERT implementation, we use the official checkpoint and PyTorch code from https://github.com/huggingface/transformers.

## B.1  Dataset Details

**OfficeHome** [64] https://www.hemanthdv.org/officeHomeDataset.html is an object recognition dataset which contains images from 4 domains. It has about 15500 images organized into 65 categories. The dataset was collected using a python web-crawler that crawled through several search engines and online image directories. The authors provided a Fair Use Notice on their website.

**VisDA-2017** [45] https://github.com/VisionLearningGroup/taskcv-2017-public/tree/master/classification uses synthetic object images rendered from CAD models as the training domain and real object images cropped from the COCO dataset as the validation domain. The authors provided a Term of Use on the website.

**DomainNet** [46] http://ai.bu.edu/M3SDA/#dataset contains images from clipart, infograph, painting, real, and sketch domains collected by searching a category name combined with a domain name from searching engines. The authors provided a Fair Use Notice on their website.

**Amazon Review** [10] https://www.cs.jhu.edu/~mdredze/datasets/sentiment/ contains product reviews taken from Amazon.com from many product types (domains). Some domains (books and dvds) have hundreds of thousands of reviews. Others (musical instruments) have only a few hundred. Reviews contain star ratings (1 to 5 stars) that can be converted into binary labels if needed.

## B.2  Bi-level Optimization

In Section 3.1, we highlight that the optimization of CST involves bi-level optimization. In the inner loop (equation 4), we train the target classifier $\theta_t(\phi)$ on top of the shared representations $\phi$, thus $\theta_t(\phi)$ is a function of $\phi$. Moreover, the target classifier $\theta_t(\phi)$ is trained with target pseudo-labels $y'$, which are the sharpened version of the outputs of the source classifier $\theta_s$ on top of the shared representations $\phi$. In this sense, $\theta_t(\phi)$ relies on $\theta_s$ and $\phi$ through $y'$ implicitly, too. In the outer loop (equation 5), we update the shared representations $\phi$ and the source classifier $\theta_s$ to make both the source classifier $\theta_s$ and the target classifier $\theta_t(\phi)$ perform well on the source domain. Since $\theta_t(\phi)$ relies on $\phi$ and $\theta_s$, the objective of equation 5 is a bi-level optimization problem. We can derive the gradient of the loss w.r.t. $\phi$ and $\theta_s$ as follows:

$$\nabla_\phi[L_{\widehat{P}}(\theta_s,\phi) + L_{\widehat{P}}(\hat{\theta}_t(\phi),\phi)] \tag{75}$$

$$=\nabla_\phi L_{\widehat{P}}(\theta_s,\phi) + \frac{\partial L_{\widehat{P}}(\hat{\theta}_t(\phi),\phi)}{\partial \phi} + \frac{\partial L_{\widehat{P}}(\hat{\theta}_t(\phi),\phi)}{\partial \hat{\theta}_t(\phi)} \frac{\mathrm{d}\hat{\theta}_t(\phi)}{\mathrm{d}\phi}$$

$$=\nabla_\phi L_{\widehat{P}}(\theta_s,\phi) + \frac{\partial L_{\widehat{P}}(\hat{\theta}_t(\phi),\phi)}{\partial \phi} + \frac{\partial L_{\widehat{P}}(\hat{\theta}_t(\phi),\phi)}{\partial \hat{\theta}_t(\phi)} \left[ \frac{\partial \hat{\theta}_t(\phi)}{\partial \phi} + \frac{\partial \hat{\theta}_t(\phi)}{\partial y'} \frac{\partial y'}{\partial \phi} \right].$$

$$\nabla_{\theta_s}[L_{\widehat{P}}(\theta_s,\phi) + L_{\widehat{P}}(\hat{\theta}_t(\phi),\phi)] = \nabla_{\theta_s} L_{\widehat{P}}(\theta_s,\phi) + \frac{\partial L_{\widehat{P}}(\hat{\theta}_t(\phi),\phi)}{\partial \hat{\theta}_t(\phi)} \frac{\partial \hat{\theta}_t(\phi)}{\partial y'} \frac{\partial y'}{\partial \theta_s}. \tag{76}$$

However, following the standard practice in self-training, we use label-sharpening to obtain target pseudo-labels $y'$, i.e. $y' = \arg\max_i\{f_{\theta_s,\phi}(x)_{[i]}\}$. Thus, $y'$ is not differentiable w.r.t. $\theta_s$ and $\phi$. We treat the gradient of $y'$ w.r.t. $\theta_s$ and $\phi$ as 0 in equation 75 and equation 76, making optimization easier. This modification leads to exactly equation 7 and equation 8 in Algorithm 1 together with the Tsallis entropy loss.

**Speeding up bi-level optimization with MSE loss.** Standard methods of bi-level optimization back-propagate through the inner loop algorithm, which requires computing the second-order derivative (Hessian-vector products) and can be unstable. We propose to use MSE loss instead of cross entropy

in the inner loop when training the head $\hat{\theta}(\phi)$ to calculate the analytical solution with least square and directly back-propagate to the outer loop without calculating second-order derivatives. The framework is as fast as training the two heads jointly. To adopt MSE loss in multi-class classification, we use the one-hot embedding as the output and train a multi-variate regressor following the protocol of Arora et al. [3]. We calculate the least square solution of $\theta_t(\phi)$ based on one minibatch following the protocol of Bertinetto et al. [9]. We also provide results of varying batchsize to verify the performance of this approximation in Table 7. Results indicate that the performance of CST is stable in a wide range of batchsizes.

Table 7: Accuracy (%) on VisDA-2017 with ResNet-50

| Method | Accuracy |
|---|---|
| CST (batchzize 32) | $79.9 \pm 0.6$ |
| CST (batchzize 64) | $79.9 \pm 0.5$ |
| CST (batchzize 128) | $79.6 \pm 0.4$ |
| CST (batchzize 256) | $79.0 \pm 0.6$ |

### B.3 Selection of $\alpha$

We can also update $\alpha$ with gradient methods auto-differentiation tools as we treat $\phi$. However, since $\alpha$ has only one parameter but many other parameters ($\theta_{s,\alpha}$ and $\theta_{t,\alpha}$) rely on it, using gradient methods is costly. To ease the computational cost, we choose to discretize the feasible region of $\alpha \in [1, 2]$ with $\alpha \in \{1.0, 1.1, \cdots 1.9, 2.0\}$, and train $\theta_{s,\alpha}$ with each $\alpha \in \{1.0, 1.1, \cdots 1.9, 2.0\}$ to generate pseudo-labels and train $\hat{\theta}_{t,\alpha}$ on pseudo-labels corresponding to each value of $\alpha$. Then we select the $\alpha \in \{1.0, 1.1, \cdots 1.9, 2.0\}$ with best performance on the source dataset following equation 10. We also update $\alpha$ at the start of each epoch, since we found more frequent update leads to no performance gain. Since we only need to select $\alpha$ once at the start of each epoch, the resulting additional computational cost only relates to training the linear head on the source and target datasets for additional 11 times per epoch, which is negligible compared to training the backbone.

We plot the change of $\alpha$ throughout training in Figure 6. $\alpha$ converges to smaller value at the end of training, indicating that the penalization on uncertainty is increasing. Also note that $\alpha$ tends decrease slower for "heuristically distant" source and target domains. This corroborates the intuition that we need to penalize uncertain predictions mildly especially when the domain gap is large.

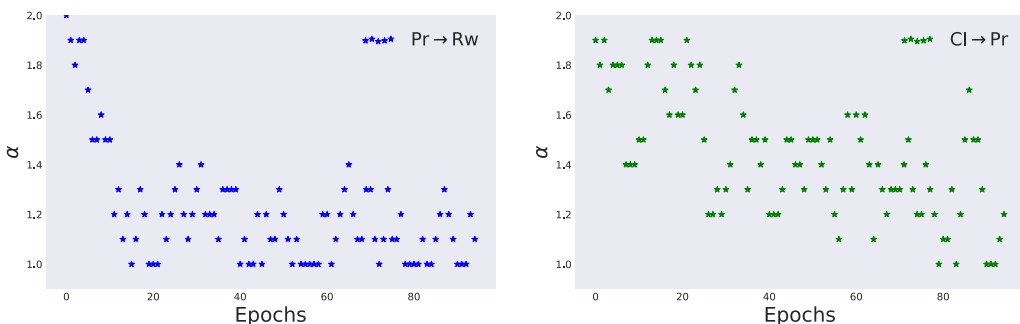

Figure 6: **Change of $\alpha$ during training.**

**Difference between $\theta_s$ and $\theta_{s,\alpha}$.** We use $\theta_s$ to update the feature extractor $\phi$ and test on the target domain after training. $\theta_{s,\alpha}$ is only used to search the optimal $\alpha$, and the gradient does not back-propagate to $\phi$.

## C  Additional Experiment Details

### C.1  Additional Details of Section 2.1

In Figure 2 (Left), we use VisDA-2017 with 12 classes. To simulate the i.i.d., covariate shift and label shift setup, we resample the original Synthetic (source) and Real (target) datasets. In the i.i.d. setting,

the labeled dataset consists of 1000 random samples per class from Real, and the unlabeled dataset consists of 1000 random samples (no overlapping with the labeled dataset) per class from Real. In the covariate shift setting, the labeled dataset consists of 1000 random samples per class from Synthetic, and the unlabeled dataset consists of 1000 random samples per class from Real. In the label shift setting, the number of examples is $[1800, 1440, 1152, 922, 737, 590, 472, 377, 302, 240, 193, 154]$ for each class of the labeled dataset and 1000 for each class of the unlabeled dataset, with both labeled and unlabeled datasets sampled randomly from Real. We train the model on labeled data until convergence to generate pseudo-labels for the unlabeled data. Then we calculate the ratio of classes in pseudo-labels and ground truth.

In Figure 2 (Middle), we also visualize the change of pseudo-label accuracy and the distance $d_{\mathrm{TV}}$ throughout standard self-training on original Synthetic and Real datasets. Here standard self-training refers to equation 2 with label-sharpening.

In both Figure 2 (Right) and this subsection, confidence refers to the maximum soft-max output value, and entropy is defined as $\sum_i -y_i \log(y_i)$. We change the confidence threshold from 0 to 1 and entropy threshold from 0 to $\log(\texttt{numclasses})$. Then we plot the point (False Positive Rate, True Positive Rate) in the plane.

In Section 2.1, we measure the quality of pseudo-labels with the total variation between pseudo-label distribution and ground-truth distribution. We show this quantity is upper-bounded by the accuracy of pseudo-labels. Intuitively, when the pseudo-label distribution and ground-truth distribution are the same, the output can still be incorrect. (e.g., in a binary problem, $P(Y=1) = P(Y=0) = 0.5$, and $\hat{Y} \sim \mathrm{uniform}[0,1]$ but is independent of $X$.) Recall that $d_{\mathrm{TV}}(Y, \hat{Y}) = \sup_{E \subset [C]} |P(Y \in E) - P(\hat{Y} \in E)|$. Suppose the supremum is reached by $\hat{E}$. Without loss of generality, assume $P(Y = c) > P(\hat{Y} = c)$ for all $c \in \hat{E}$. Then $P(Y = c) \leq P(\hat{Y} = c)$ for all $c \notin \hat{E}$.

$$
\begin{aligned}
P(Y \neq \hat{Y}) &= \sum_{c=1}^{C} P(Y = c, \ \hat{Y} \neq c) \\
&= \sum_{c \in \hat{E}} P(Y = c, \ \hat{Y} \neq c) + \sum_{c \notin \hat{E}} P(Y = c, \ \hat{Y} \neq c) \\
&\geq \sum_{c \in \hat{E}} P(Y = c) - P(\hat{Y} = c) \\
&= d_{\mathrm{TV}}(Y, \hat{Y}).
\end{aligned}
$$

In the equality, we use $P(Y = c) > P(\hat{Y} = c)$ when $c \in \hat{E}$, so $P(Y = c, \ \hat{Y} \neq c) \geq P(Y = c) - P(\hat{Y} = c)$ if $c \in \hat{E}$. Also note that $P(Y = c, \ \hat{Y} \neq c) \geq 0$ if $c \notin \hat{E}$.

In this subsection, we provide additional results of Section 2.1. We visualize the distributions of pseudo-labels and ground-truth with ResNet-50 backbones on Art→Clipart, Product→Art, Clipart→Real World, Art→Real World and Real World →Product tasks (without resampling) in Figures 7, 8, 9, 10, and 11 respectively. We also visualize the ROC curve of pseudo-label selection with confidence threshold. Results on Art→Clipart, Product→Art, Clipart→Real World, Art→Real World and Real World →Product are similar to VisDA-2017. When the pseudo-labels are generated from models trained on different distributions, they can become especially unreliable in that examples of several classes are almost misclassified into other classes. Domain shift also makes the selection of correct pseudo-labels more difficult than standard semi-supervised learning.

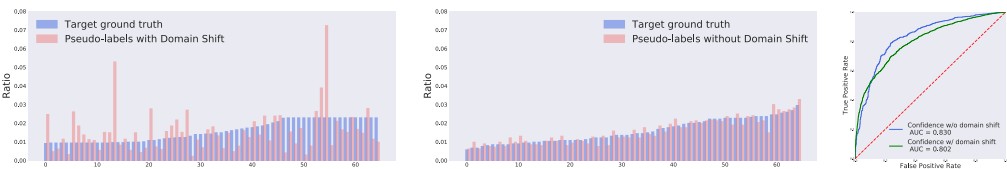

Figure 7: **Analysis of pseudo-labels under domain shift on Art→Clipart.** Left: Comparison of pseudo-label distributions with and without domain shift. Right: Comparison of pseudo-label selection with and without domain shift.

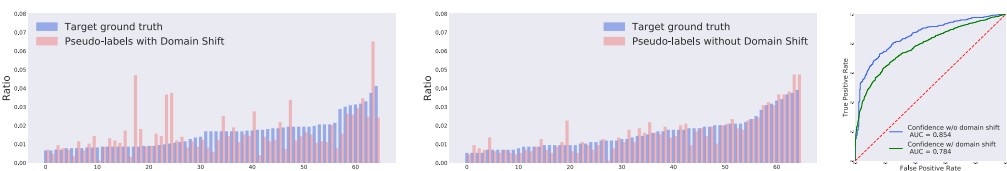

Figure 8: **Analysis of pseudo-labels under domain shift on Product→Art.** Left: Comparison of pseudo-label distributions with and without domain shift. Right: Comparison of pseudo-label selection with and without domain shift.

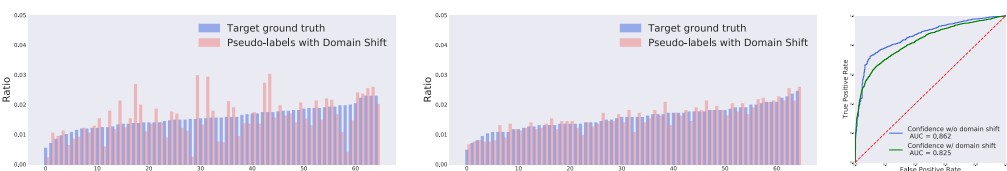

Figure 9: **Analysis of pseudo-labels under domain shift on Clipart→Real World.** Left: Comparison of pseudo-label distributions with and without domain shift. Right: Comparison of pseudo-label selection with and without domain shift.

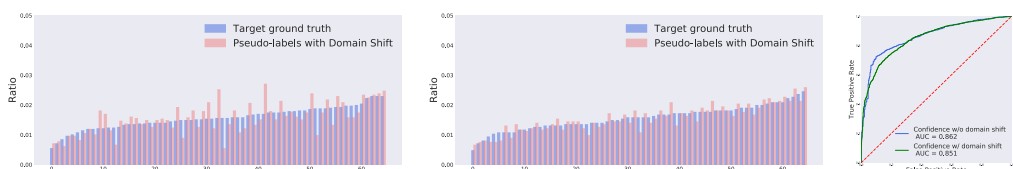

Figure 10: **Analysis of pseudo-labels under domain shift on Art→Real World.** Left: Comparison of pseudo-label distributions with and without domain shift. Right: Comparison of pseudo-label selection with and without domain shift.

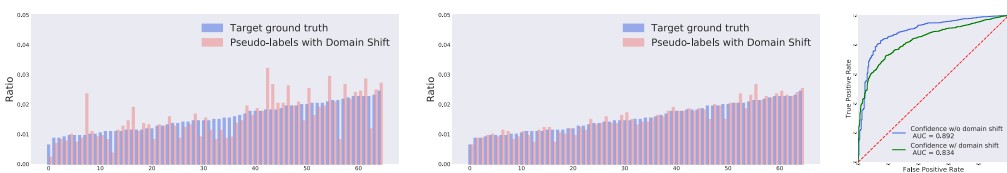

Figure 11: **Analysis of pseudo-labels under domain shift on Real World→Product.** Left: Comparison of pseudo-label distributions with and without domain shift. Right: Comparison of pseudo-label selection with and without domain shift.

## C.2   Results on digit datasets

We provide results on the digit datasets to test the performance of the proposed method without using pre-training. We use DTN architecture following Long et al. [37]. Results in Table 8 indicate that CST achieve comparable performance to state-of-the-art.

Table 8: Accuracy (%) on digits datasets with DTN

| Method | MNIST→USPS | SVHN→MNIST |
|---|---|---|
| CDAN | $95.6 \pm 0.2$ | $96.9 \pm 0.2$ |
| RWOT (CVPR 2020) | $98.5 \pm 0.2$ | $97.5 \pm 0.2$ |
| **CST** | $98.5 \pm 0.2$ | $\mathbf{98.2} \pm 0.2$ |

## C.3   Results on DomainNet

We test the performance of the proposed method on the 40-class DomainNet [46] subset following the protocol of Tan et al. [60]. Results in Table 9 indicate that CST outperforms MDD by a large margin.

Table 9: Accuracy (%) on DomainNet for unsupervised domain adaptation (`ResNet-50`).

| Method | R-C | R-P | R-S | C-R | C-P | C-S | P-R | P-C | P-S | S-R | S-C | S-P | Avg. |
|---|---|---|---|---|---|---|---|---|---|---|---|---|---|
| DANN [22] | 63.4 | 73.6 | 72.6 | 86.5 | 65.7 | 70.6 | 86.9 | 73.2 | 70.2 | 85.7 | 75.2 | 70.0 | 74.5 |
| COAL [60] | 73.9 | 75.4 | 70.5 | 89.6 | 70.0 | 71.3 | 89.8 | 68.0 | 70.5 | 88.0 | 73.2 | 70.5 | 75.9 |
| MDD [73] | 77.6 | 75.7 | 74.2 | 89.5 | 74.2 | 75.6 | 90.2 | 76.0 | 74.6 | 86.7 | 72.9 | 73.2 | 78.4 |
| **CST** | **83.9** | **78.1** | **77.5** | **90.9** | **76.4** | **79.7** | **90.8** | **82.5** | **76.5** | **90.0** | **82.8** | **74.4** | **82.0** |

## C.4   Standard deviations of Tables

We visualize the performance of CST and best baselines in Table 3 with standard deviations. Results indicate that the improvement of CST over previous methods is significant.

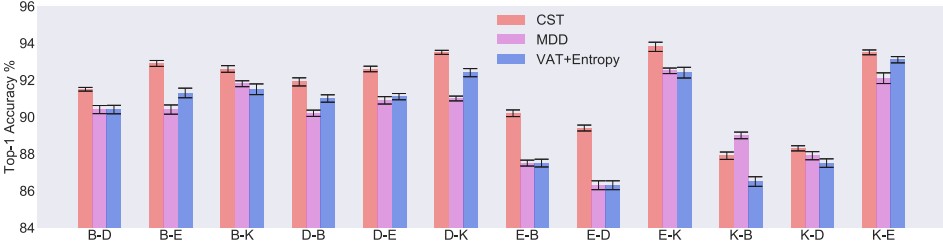

Figure 12: **Visualization of standard deviations of CST and baselines.** CST outperforms baselines significantly on all tasks except K→B.

## D   Limitations of CST and Future Directions

CST overcomes the drawbacks of standard pseudo-labeling in domain adaptation by dealing with the domain discrepancy explicitly with the cycle loss. However, pseudo-labeling is only one main direction of semi-supervised learning. Consistency regularization [40] and self-ensembling [4] are also important paradigms in semi-supervised learning. How to apply them to the setting with distribution shift and achieve consistent performance gain is still an open question. More recently, Carlini [12] investigated the effect of adversarial unlabeled data poisoning to self-training. Future works can extend CST to this setting and extend consistency regularization as a potential way of defense.

## E   Broader Impact

This work studies and improves self-training in the unsupervised domain adaptation setting. When deployed in real-world applications, distribution shift between labeled and unlabeled data can come in various ways. Although the quality of labeled datasets can be monitored, enabling the mitigation of bias in pre-processing, bias in unlabeled datasets can be intractable. Self-training with biased unlabeled data is highly risky since it may potentially amplify the biased models predictions. This work explores how to mitigate the effect of dataset bias in unlabeled data, and can potentially promote fair self-training systems.