# OpenReview forum: "Cycle Self-Training for Domain Adaptation"
_NeurIPS.cc/2021/Conference — NeurIPS 2021 Poster_

### Official Review · Reviewer_Yv3B · 2021-07-16

**Rating:** 7
**Confidence:** 4

**Summary:**

The paper tackles the problem of domain adaptation with a newly proposed cycle self-training algorithm. Given observed that the pseudo labels are noisy and existing de-noising methods require ad-hoc hyper-parameters for specific tasks, the authors attempt to progressively refine the pseudo labels by the capability of the network itself. With the intuition of transferring the source-domain knowledge to the target domain, the authors propose to regularize the pseudo labels with a cyclic training pipeline. Tsallis Entropy is further introduced to improve the label quality. Experiments on both visual classification and linguistic sentiment classification indicate the effectiveness of the introduced method.

**Limitations And Societal Impact:**

The introduced cycle self-training could only be applied when two domains share the same label set. How to properly generalize this method to real-world scenarios (e.g., open-set) is important.

**Main Review:**

Strengths:

+ The paper is well written and easy to follow. The figures and charts are nice and help to understand.

+ The analysis of limitations of existing self-training methods in Sec. 2.1 well motivates the introduce method.

+ The authors carefully analyze the introduced method both theoretically and empirically.

+ The introduced method is simple but effective. It seems easy to implement and I believe the results can be well reproduced.

+ The introduced Tsallis Entropy can be seamlessly integrated into existing methods, e.g., FixMatch, and achieved significant performance gains as shown in Table 5.


I am overall satisfied with this paper and would like to give some comments that may make this paper better.

1. The notations are a bit complex.

2. Despite that the introduced method is very simple, it is also necessary to compare with more SoTA methods (in CVPR2020/ECCV2020/…) in the experimental part (Table 2/3/…).

3. Generalizing this method to open-set domain adaptation would make it more applicable in real-world scenarios.


After rebuttal: My concerns have been well addressed. I recommend accepting this paper.

**Time Spent Reviewing:**

4

---

> ### Author Response · Authors · 2021-08-09
> **Response to Reviewer Yv3B**
>
> Thank you for providing the constructive comments! We will answer them as below.
>
> **Q1:** Despite that the introduced method is very simple, it is also necessary to compare with more SoTA methods (in CVPR2020/ECCV2020/…) in the experimental part (Table 2/3/…).
>
> We aim at proposing a **strong and simple method** that can serve as a base backbone in more complicated method and work under both computer vision and language tasks. Here, to make it comparable with the mentioned SoTA methods, we also provide CST + mixup, which combines CST with mixup (a common computer vision technique widely used by the SoTA methods). We provide comparison of accuracy averaged over classes on VisDA 2017 in the Table below:
>
> | Method | Accuracy averaged over classes |
> | :-- | :--: |
> | MDD (ICML 2019) [5]  | 80.4 |
> | MCC (ECCV 2020) [7] | 82.2 |
> | STAR (CVPR 2020) [6] | 82.7 |
> | FixBi (CVPR 2021) [2] | 87.2 |
> | CST | 84.0 |
> | CST + mixup | 86.7 |
>
> As can be seen, by enhancing our simple method with a common trick (mixup), it can already perform on par with the much more complex SoTA methods such as [1] and [2]. Note that [1] and [2] include temperature calibration, iterative clustering, class-wise sampling and mixup, and threshold target samples to select pseudo-labels. We will make our evaluation more complete by adding the above results.
>
> **Q2:** Generalizing this method to open-set domain adaptation would make it more applicable in real-world scenarios.
>
> We recognize that open-set domain adaptation is important in real-world applications. However, the aim of this paper is to propose a principled solution to standard domain adaptation setting which can be further incorporated as backbones into other settings (such as open-set DA and continual DA). To address the concern, we extend CST by incorporating confidence thresholding (CST + threshold) to distinguish closed-set classes from open-set classes. At inference, we reject the examples below the confidence threshold as "unknown" classes and further classify target examples in the closed-set classes. Results of open-set domain adaptation on VisDA-2017 are provided in the table below.
>
> | Method | OS |
> | :-- | :--: |
> | OSBP [4] | 62.9 |
> | UAN [3] | 66.0 |
> | CST + threshold | 66.5 |
>
> Here OS means the normalized accuracy for all classes including the "unknown" as one class.
>
>
>
> [1] Contrastive Adaptation Network for Unsupervised Domain Adaptation. CVPR 2019.
> [2] FixBi: Bridging Domain Spaces for Unsupervised Domain Adaptation. CVPR 2021.
> [3] Universal Domain Adaptation. CVPR 2019.
> [4] Open-set Domain Adaptation by Backpropagation. CVPR 2018.
> [5] Bridging Theory and Algorithm for Domain Adaptation. ICML 2019.
> [6] Stochastic Classifiers for Unsupervised Domain Adaptation. CVPR 2020.
> [7] Minimum Class Confusion for Versatile Domain Adaptation. ECCV 2020.

---

### Official Review · Reviewer_PLkj · 2021-07-16

**Rating:** 6
**Confidence:** 5

**Summary:**

This paper presents a novel approach for unsupervised domain adaptation (UDA) based on cycle self-training. Instead of conventional self-training, there are a generated and two attached classifiers with different purposes. The source classifier, mainly trained on the source loss, is applied to infer pseudo target labels. The target classifier is trained by the self-training loss based on the pseudo labels provided by the source classifier.  The authors also propose the Tsallis entropy as the regularization term, which can be seen as a side contribution.

**Limitations And Societal Impact:**

Yes. The authors provide adequate discussion of limitations and potential negative societal impact.

**Main Review:**

Strengths:

+ The paper is organized in clear logic, which is easier to read. The discussion of related works is comprehensive enough to cover most of the topics in the field.

+ There is a detailed analysis of self-training in different cases: with or without domain shift, distribution change, etc. It is helpful to deeply understand the effects of pseudo labels in the DA setting.

+ Tsallis entropy is an interesting idea that seems to fit better with the DA problem than the conventional entropy loss. And the discussion of its motivation is comprehensive and clear.

Weaknesses:

-The cycle self-training seems not novel and too simple. Some previous works on the tri-training-based DA have tried similar ideas where a fixing classifier is used to help another classifier. And I am not quite convinced by the design of the source classifier, which seems not to be trained by the target samples. If so, it is hard for me to believe that it can infer the accurate target pseudo labels without seeing them. Will the pseudo labels have biased towards the source domain? How to address such a problem.

-The experimental results are not strong enough. Although it outperforms the SOTA SSL methods and some baseline UDA methods, some ignored UDA works such as [1] and [2] can beat the proposed method by a large margin.



Other Comments:

1. The right subfigure of Fig2 is too small to read. It will be much better if enlarging it.

2. Please provide which dataset is used in the caption of Fig 2.

3. In lines 36 – 42, the motivation of the reverse step is not convincing enough. The target pseudo labels informative to the source domain may also be harmful. The authors should provide more solid experimental results to support such a claim.

[1] Contrastive Adaptation Network for Unsupervised Domain Adaptation. CVPR 2019.
[2] FixBi: Bridging Domain Spaces for Unsupervised Domain Adaptation. CVPR 2021.

After rebuttal:
My main concerns are addressed. I'd like to increase my rating score. Thanks for the detailed feedback.

**Time Spent Reviewing:**

14

---

> ### Author Response · Authors · 2021-08-09
> **Response to Reviewer PLkj**
>
> Thanks for taking the time reading our paper and providing detailed comments. We will fully address the concerns below.
>
> **Q1:** The cycle self-training seems not novel and too simple. Previous works on the tri-training-based DA have tried similar ideas where a fixing classifier is used to help another classifier.
>
> As agreed unanimously by Reviewers wNMh, txSFT and Yv3B, the proposed cycle self-training (CST) is well motivated and technically novel. We further elaborate how our method is different essentially from tri-training or adversarial feature adaptation:
> - Tri-training such as ADTA [3] selects pseudo-labels with high agreement between auxiliary classifiers and confidence threshold, which improves upon standard self-training.
> - Adversarial feature adaptation methods such as MCD [4] and MDD [5] train an auxiliary classifier to adversarially disagree with the source classifier to approximate the $H\Delta H$ divergence [6] and then minimize it.
> - Although these methods also use multiple classifiers, they fall under either **standard self-training** or **adversarial feature adaptation**, which we have shown potential limitations in $\underline{\text{Theorems 3 and 4}}$.
> - In contrast, the mechanism of the proposed cycle self-training is distinct in that it harnesses the novel **reverse step** to improve the quality of pseudo-labels in a principled manner. **Our method also has completely different theoretical grounds**, which we presented in $\underline{\text{Theorems 1 and 2}}$.
>
> We will cite the mentioned references and make the comparison clearer in the revision.
>
> **Q2:** I am not quite convinced by the design of the source classifier, which seems not to be trained by the target samples. Will the pseudo labels have biased towards the source domain?
>
> Sorry for the confusion. We clarify here that the source classifier is indeed trained with target examples in two ways:
> -  In $\underline{\text{Eqn. 11}}$, we train the source classifier with the Tsallis entropy to adaptively regularize its confidence on the target examples.
> -  In $\underline{\text{Eqn. 4}}$, the target classifier is trained on the target examples with pseudo-labels generated by the source classifier, and the shared feature extractor is updated to make the target classifier perform well on the source domain in $\underline{\text{Eqn. 5}}$, which implicitly refines the outputs of the source classifier on the target examples.
>
> Thereby, as training proceeds, both the accuracy of the target classifier on the source domain and the accuracy of the source classifier on the target domain increase as illustrated in $\underline{\text{Figure 5}}$ (Left). The effectiveness of the source classifier is also justified theoretically in $\underline{\text{Section 4}}$. When the distribution shift satisfies the **expansion assumption** (Definition 1), the source classifier is guaranteed to perform well on the target domain and will not be biased towards the source domain.
>
> **Q3:** The experimental results are not strong enough. Some UDA works such as [1] and [2] can beat the proposed method by a large margin.
>
> First, the goal of this work is to propose a **simple and principled method with theoretical guarantee** which can work for both computer vision and NLP tasks (Results on Amazon Review presented in $\underline{\text{Table 3}}$ and reiterated in the table below). Note that [1] and [2] are tailored to computer vision tasks which include specific techniques such as *temperature calibration, class-wise sampling, iterative clustering and mixup*. Both of them threshold target samples to select pseudo-labels, which needs extensive tuning of the confidence threshold for new tasks.
>
> | Method | Average Acc |
> | :-- | :--: |
> | BERT | 89.2 |
> | DANN | 89.9 |
> | VAT | 90.1 |
> | VAT + Entropy | 90.1 |
> | MDD | 90.0 |
> | CST | 91.5 |
>
> Second, the evaluation metric provided in [1] and [2] on VisDA is **per-class accuracy**, while we provide results of **per-instance accuracy** following [5]. In the Table below, we further compare CST with the state-of-the-arts in the computer vision literature based on accuracy averaged per-class. Note that some of these methods may be complex with many technical tricks. However, to still keep it simple, we only **incorporate the mixup** (a common technique in computer vision) to our CST to formulate CST + mixup, which can already perform comparably well with [1] and [2].
>
> | Method | Accuracy averaged over classes |
> | :-- | :--: |
> | MDD (ICML 2019) [5]  | 80.4 |
> | MCC (ECCV 2020) [8] | 82.2 |
> | STAR (CVPR 2020) [7] | 82.7 |
> | FixBi (CVPR 2021) [2] | 87.2 |
> | CST | 84.0 |
> | CST + mixup | 86.7 |
>
> [1] Contrastive Adaptation Network for Unsupervised Domain Adaptation. CVPR 2019.
> [2] FixBi: Bridging Domain Spaces for Unsupervised Domain Adaptation. CVPR 2021.
> [3] Asymmetric Tri-training for Unsupervised Domain Adaptation. ICML 2017.
> [4] Maximum Classifier Discrepancy for Unsupervised Domain Adaptation. CVPR 2018.
> [5] Bridging Theory and Algorithm for Domain Adaptation. ICML 2019.
> [6] A Theory of Learning from Different Domains. Machine Learning 2010.
> [7] Stochastic Classifiers for Unsupervised Domain Adaptation. CVPR 2020.
> [8] Minimum Class Confusion for Versatile Domain Adaptation. ECCV 2020.

---

> > ### Author Response · Authors · 2021-08-18
> > **Reviewer feedback to author response**
> >
> > Dear Reviewer PLkj,
> >
> > Many thanks for your time and efforts in reviewing our paper.
> >
> > We kindly remind that we are less than one week into the discussion period. We have made extensive effort to try to successfully address your concerns and answer your questions, by providing supporting experiments you requested and highlighting some results originally presented in the $\underline{\text{main paper}}$​ and the $\underline{\text{supplementary material}}$​.
> >
> > If you have any further concerns or questions, please do not hesitate to let us know, and we will respond to them timely.
> >
> > All the best,
> > Authors

---

> > ### Comment · Reviewer_PLkj · 2021-08-19
> > **My main concerns are addressed**
> >
> > Thanks for the detailed and well-organized responses. My main concerns have been addressed. I'd like to increase my rating score.

---

### Official Review · Reviewer_txSf · 2021-07-17

**Rating:** 7
**Confidence:** 4

**Summary:**

Proposes cycle self-training (CST) an unsupervised domain adaptation (UDA) algorithm that cycles between standard self-training of a target classifier on (source model generated) target pseudolabels, and a new cycle self-training step that updates backbone representations so as to generalize the target classifier to the source domain. An additional self-training objective based on the Tsallis entropy is introduced. Theoretical and empirical results are presented on several standard UDA benchmarks.

**Limitations And Societal Impact:**

Authors adequately addressed the limitations and potential negative societal impact.

**Main Review:**

Originality

The proposed approach is novel and original.

Quality

– The paper does well to analyze and clearly describe some of the shortcomings of self-training (Sec 2.1).

– The proposed approach, although slightly complex, is well-designed, explained, and well-supported by theoretical analysis (Sec 4).

– The performance gains of the proposed approach are consistent and promising, and the paper does well to present results on several benchmarks (including additional ones in the supplement) across vision and language tasks, and compare to a wide range of baselines, including strong baselines derived from combining existing methods.

Clarity

– The paper is well-written and easy to follow.

– [minor] The flow of logic in Figure 1 is somewhat difficult to follow and it took me multiple reads. It would be good to try and simplify it.

– [minor] In Eq 3-5, it would be good to include what instances (source / target) the objective operates on

Significance

The paper focuses on the UDA problem that is of considerable research interest and the proposed approach improves upon prior work in a wide variety of settings.

Overall

This is a well-written paper that proposes an interesting, effective, and principled UDA approach. I recommend acceptance.

---Post-Rebuttal----

I have read the author response and other reviews, and would like to keep my accept rating.

**Time Spent Reviewing:**

3

---

> ### Author Response · Authors · 2021-08-09
> **Response to Reviewer txSf**
>
> Thanks for your constructive suggestions!
>
> We will try to simplify Figure 1 and make the comparison between self-training and cycle self-training clearer. In $\underline{\text{Eqn. 3-5}}$, we already stated the domain that the objectives operate on ($x \sim \widehat{P}$ or $L_{\widehat{Q}}$). We will also make the notations clearer to avoid potential confusion.

---

### Official Review · Reviewer_wNMh · 2021-07-24

**Rating:** 8
**Confidence:** 4

**Summary:**

This paper proposes an unsupervised domain adaptation method based on cyclical self-training. There are two main components to the method: 1. a cyclical self-training algorithm that solves a bilevel optimization problem based on an in the inner loop which trains a target classifier with target pseudo-labels  and an outer loop that makes the target classifier perform well on the source domain by updating the shared representations in the outer loop. 2. an uncertainty measure based on the Tsallis entropy, which adaptively minimizes the uncertainty in target pseudo-labels, which replaces the standard Gibbs entropy used by self-training methods. Experiments on computer vision and NLP datasets show that proposed method outperforms baseline methods by a decent margin. Ablation studies show that both components of the algorithm---cyclical self-training and Tsallis entropy---meaningfully contribute to performance improvement. Qualitative and quantitative studies show that target pseudo-label quality is improved by the proposed approach.

**Limitations And Societal Impact:**

See above.

**Main Review:**

Strengths:
1. The paper is well-written and organized. The introduction section motivates the problem with current self-training methods using qualitative and quantitative experiments. The proposed approach is supported by theoretical understanding. Experiments and ablation studies help understand how the algorithm works and quantify performance improvement.

2. Experiments across data domains show good perfomance improvement over state-of-the-art methods, including those tailored to specific domains or dataset types.

Weakness:
This is not a weakness per se, but the paper would benefit from more analysis of what type of performance improvement is obtained by using this method. Understanding how label shift and covariate shift is tackled and what type of examples are classified better by this method can help spur future research in this problem space.

**Time Spent Reviewing:**

3

---

> ### Author Response · Authors · 2021-08-09
> **Response to Reviewer wNMh**
>
> Thank you for appreciating our work!
>
> We believe that understanding how label shift and covariate shift are tackled is an important way to delve into why the proposed method works well. Here we further elaborate on them through our theoretical results:
> - As informed by $\underline{\text{Theorems 3 and 4}}$, when the **label shift** makes the distributional distance of "good" features larger than that of spurious features, feature adaptation will prefer spurious features. The proposed method uses cycle self-training to relax the constraint of distributional alignment and may still succeed in the case where label shift causes feature adaptation to fail.
> - As to **covariate shift**, $\underline{\text{Theorems 1 and 2}}$ indicate that when the shift satisfies the expansion assumption ($\underline{\text{Assumption 1}}$), the proposed cycle self-training will successfully facilitate label propagation from the source to the target, thereby enabling knowledge transfer between the source and target domains.
>
> We will also provide illustrations of what type of examples that CST classifies better in the revision.

---

### Decision · Program_Chairs · 2021-09-27

**Decision:**

Accept (Poster)

**Comment:**

The reviewers unanimously agree that the paper should be accepted. I find the idea of the paper interesting and the empirical results seem encouraging. I am not convinced that the presented algorithm in its full complexity can be theoretically studied. The experimental results are encouraging but still falling behind the state-of-the-art. Notation is cluttered and exposition is not as clear as it should be. I recommend acceptance as a poster, but please incorporate reviewers' feedback to improve the paper.